# SkillEvo: An Experience Learning Framework with Reinforcement Learning for Skill Evolution

## Abstract

Large Language Models (LLMs) have evolved into agents capable of perception, reasoning, and acting in open environments. Yet, in long-horizon tasks with sparse rewards, existing methods are often inefficient. Group-based reinforcement learning (e.g., GRPO) provides critic-free and stable optimization, but its coarse credit signals cannot distinguish high-quality trajectories from those that merely succeed but contain redundant or invalid actions, leading to weak generalization. We propose **SkillEvo**(**Skill Evo**lution), a two-stage framework for efficient and sustainable agent learning. In the first stage, WebGRPO integrates a Reasoning and Execution Reward Model (RXERM) to deliver fine-grained feedback, and employs a dual-uncertainty filtering strategy to select informative tasks, improving sample efficiency and stability. In the second stage, SkillGenesis transforms trajectories into reusable skills, organized in a dynamically evolving Skill Path Graph (SPG). This enables skill composition, reuse, and the emergence of composite skills for long-term adaptability. On WebArena-Lite, SkillEvo raises the success rate of Llama-3.1-8B from 4.8% to 60.4% and GLM-4-9B from 6.1% to 57.6%, achieving new state-of-the-art results. These findings highlight that effective long-horizon learning requires not only refined credit signals but also systematic mechanisms for skill evolution.

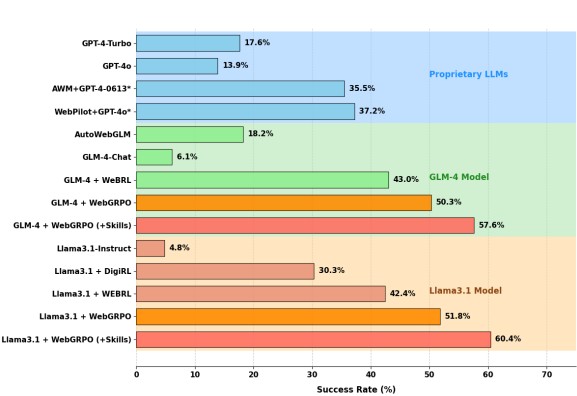

Figure 1: Comparative Success Rates of proprietary LLMs and open-sourced LLMs on WebArena-Lite.

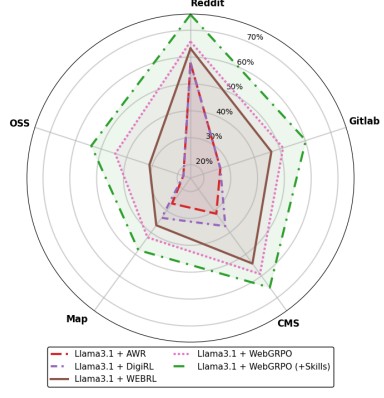

Figure 2: Comparison of Llama3.1 trained with WebGRPO and baseline methods.

## 1 Introduction

Large Language Models (LLMs)  (Team et al., 2025; OpenAI et al., 2024; Qwen et al., 2025; DeepSeek-AI et al., 2025) have rapidly advanced in recent years, evolving from static question-answering systems into versatile agents capable of perception, reasoning, and acting in open environments. In complex scenarios such as embodied navigation (Shridhar et al., 2021; Li et al., 2025; Wang et al., 2023), web interaction (Furuta et al., 2024; Zheng et al., 2024; Gou et al., 2025), and

tool-augmented reasoning (Yao et al., 2023b; Schick et al., 2023), LLMs have demonstrated strong capabilities in task planning and logical reasoning. However, enabling LLM-based agents to learn efficiently in environments that involve long horizons, multi-turn interactions, sparse rewards, and dynamic structures remains a fundamental challenge.

Recently, group-based reinforcement learning (group-based RL) has gained significant attention in large-model training. Representative methods such as RLOO (Ahmadian et al., 2024) and GRPO (Shao et al., 2024b) compute relative advantages within a batch of rollouts, thereby avoiding value function estimation. These methods exhibit desirable properties such as being critic-free, memory-efficient, and stable in convergence, and have thus become important approaches for LLM post-training. They have achieved strong performance in tasks with immediate rewards, such as mathematical reasoning (Shao et al., 2024b) and code generation (Chen et al., 2021). However, when applied directly to long-horizon environments, group-based RL faces inherent limitations: the differences among trajectories are often flattened, making it difficult to distinguish high-quality trajectories from those that contain redundant or invalid actions—even if both are labeled as "successful." This coarse-grained credit signal significantly limits the effectiveness of group-based RL in long-horizon tasks.

In fact, a successful trajectory is not necessarily a high-quality trajectory. Under the same task and initial conditions, agents may complete the task through various trajectories, some of which involve repetitive or invalid actions (e.g., frequently switching between the same webpages or revisiting the same rooms in embodied settings). If training relies solely on terminal rewards, these trajectories are treated as equally successful, thereby obscuring the value of efficient reasoning and effective execution. Over time, this leads to slower convergence and weaker generalization.

To address these challenges, we propose the SkillEvo framework, whose central insight is: optimizing long-horizon tasks requires not only finer-grained reward signals but also answers to two higher-level questions—what experiences are worth learning, and how can these experiences be consolidated into long-term capabilities?

In the first stage, SkillEvo introduces the WebGRPO module, which combines a Reasoning and Execution Reward Model (RXERM) for trajectory optimization. Unlike traditional group-based RL approaches that rely solely on overall returns, RXERM explicitly models reasoning validity and execution efficiency, thereby distinguishing high-quality from low-quality trajectories and providing more informative signals for long-horizon optimization. In addition, we propose a dual-uncertainty active learning strategy, which jointly models execution uncertainty and reasoning uncertainty to dynamically select informative instances. This strategy not only avoids overtraining on "too easy" or "too hard" cases but also naturally forms a curriculum-like schedule: ensuring efficiency within each phase while gradually increasing task difficulty across phases, thereby mitigating inefficiency and instability in long-horizon optimization (Settles, 2009).

Nevertheless, trajectory-level optimization alone is insufficient for long-term agent evolution. Even if an agent learns high-quality experiences through refined reward signals and selective training, such experiences may not be consolidated or reused, leaving the agent to start from scratch when facing new tasks. To overcome this limitation, we introduce the SkillGenesis framework as the second stage of SkillEvo. SkillGenesis systematically transforms interaction experiences into reusable skills through a three-stage process of skill proposal, skill generation, and skill evolution. These skills are organized within a dynamically expanding Skill Path Graph (SPG), which supports dynamic composition and reuse of skills, and fosters the emergence of composite skills. This mechanism enables the agent to continually expand its skill repertoire and develop adaptive, transferable capabilities across tasks. The transition from merely "completing tasks" to genuinely "acquiring and evolving skills" represents the fundamental distinction of SkillEvo compared to prior approaches.

Building on this two-stage design, SkillEvo demonstrates both theoretical and empirical advantages. Experimental results on the WebArena-Lite benchmark (Liu et al., 2024) show that our method significantly boosts task success rates: Llama-3.1-8B improves from 4.8% to 60.4%, while GLM-4-9B improves from 6.1% to 57.6%. Across both open-source and proprietary LLMs, SkillEvo consistently outperforms existing methods, achieving new state-of-the-art (SOTA) performance. These results validate our core insight: by focusing on high-quality experiences through reward modeling and uncertainty-driven selection, and further consolidating these experiences into evolving skills, LLM agents can achieve more efficient, stable, and sustainable learning in complex environments.

## 2 RELATED WORK

**Reinforcement Learning for LLM Agents.** Reinforcement learning plays a central role in enhancing reasoning and decision-making for LLM agents. Classical methods such as PPO (Schulman et al., 2017) and AWR (Peng et al., 2019a) are widely used in RLHF, but their reliance on value networks incurs significant computational overhead and makes effective credit assignment difficult in long-horizon tasks. Recently, group-based reinforcement learning methods (e.g., GRPO (Shao et al., 2024a), Dr.GRPO (Liu et al., 2025), DAPO (Yu et al., 2025)) have gained attention for avoiding value modeling by computing relative advantages within groups of trajectories. These approaches achieve critic-free, efficient, and stable optimization, and have shown strong performance in short-horizon tasks. However, in long-horizon environments with sparse and delayed rewards, such methods often provide only coarse-grained credit signals, failing to distinguish high-quality trajectories from those containing redundant actions. To overcome this limitation, we propose the WebGRPO module, which integrates a Reasoning and Execution Reward Model (RXERM) to deliver fine-grained feedback, and further design a dual-uncertainty active filtering mechanism to dynamically select informative task instances. This approach improves sample efficiency and training stability, addressing the challenges of sparse rewards and insufficient credit assignment in long-horizon optimization.

**LLMs as Agents.** In recent years, large language models (LLMs) have increasingly been employed as agents in open environments. Prompt-based approaches (e.g., ReAct (Yao et al., 2023a) and Reflexion (Shinn et al., 2023)) leverage structured reasoning prompts and tool use (Schick et al., 2023; Xie et al., 2024) to extend model interaction capabilities, but they exhibit limited long-term learning and generalization. Training-based methods instead learn directly from interaction trajectories via behavior cloning or reinforcement learning, achieving stronger task grounding and execution robustness (Shen et al., 2024; Lai et al., 2024). Meanwhile, several works have explored skill abstraction and reuse (Zheng et al., 2025; Wu et al., 2025) to improve cross-task adaptability. However, these methods generally lack systematic mechanisms for skill evolution, making it difficult to continuously expand capabilities. To address this limitation, we introduce the SkillGenesis framework, which transforms interaction experiences into reusable skills through proposal, generation, and evolution stages, and organizes them via a Skill Path Graph (SPG). This design supports dynamic reuse and the emergence of composite skills, enabling long-term consolidation of capabilities.

## 3 METHODOLOGY

### 3.1 THE FINITE-HORIZON MARKOV DECISION PROCESS FOR WEB TASK

We model the process of completing a Web task as a finite-horizon Markov Decision Process (MDP), denoted as $\mathcal{M} = (T, S, A, P, K)$, where $T$ represents the task goals; $S$ represents the state space (e.g., observation sequences or interaction histories); $A$ represents the action space, including all executable operations of the agent (e.g., `<Click>`, `<Type>`, `<Search>`, etc.); $P$ represents the state transition dynamics and reward generation process; and $K$ denotes the maximum interaction steps, after which the task terminates. Given a task goal $T$, the agent needs to complete the corresponding task. The agent policy $\pi_\theta$ generates an action $a_t$ based on the task goal $T$, the current state $s_t$, and the interaction history $\tau_{<t} = \{s_0, a_0, r_0, \ldots, s_{t-1}, a_{t-1}, r_{t-1}\}$:

$$a_t \sim \pi_\theta(\cdot|T, s_t, \tau_{<t}), \quad (r_t, s_{t+1}) \sim P(\cdot|T, s_t, a_t) \tag{1}$$

The agent receives a positive reward only if the task is completed. In the finite-horizon setting, the trajectory ends upon task completion or reaching $K$ steps. The final trajectory $\tau$ is as follows:

$$\tau = \{s_0, a_0, r_0, s_1, a_1, r_1, \ldots, s_K\} \tag{2}$$

### 3.2 TRAJECTORY-LEVEL OPTIMIZATION FOR WEB INTERACTION POLICIES

In web environments, tasks involve complex structures, long action sequences, and sparse, delayed rewards. To address this, WebGRPO optimizes the entire reasoning–interaction trajectory, enhancing long-term reasoning and policy execution. Unlike existing methods (Qi et al., 2025; Lai et al., 2024), WebGRPO directly maximizes cumulative rewards by optimizing full trajectories (including observations, reasoning processes, actions and feedback), as illustrated in Figure 3.

$$J_{\text{WebGRPO}}(\theta) = \mathbb{E}_{M, \tau \sim \pi_\theta}[R(\tau)] \tag{3}$$

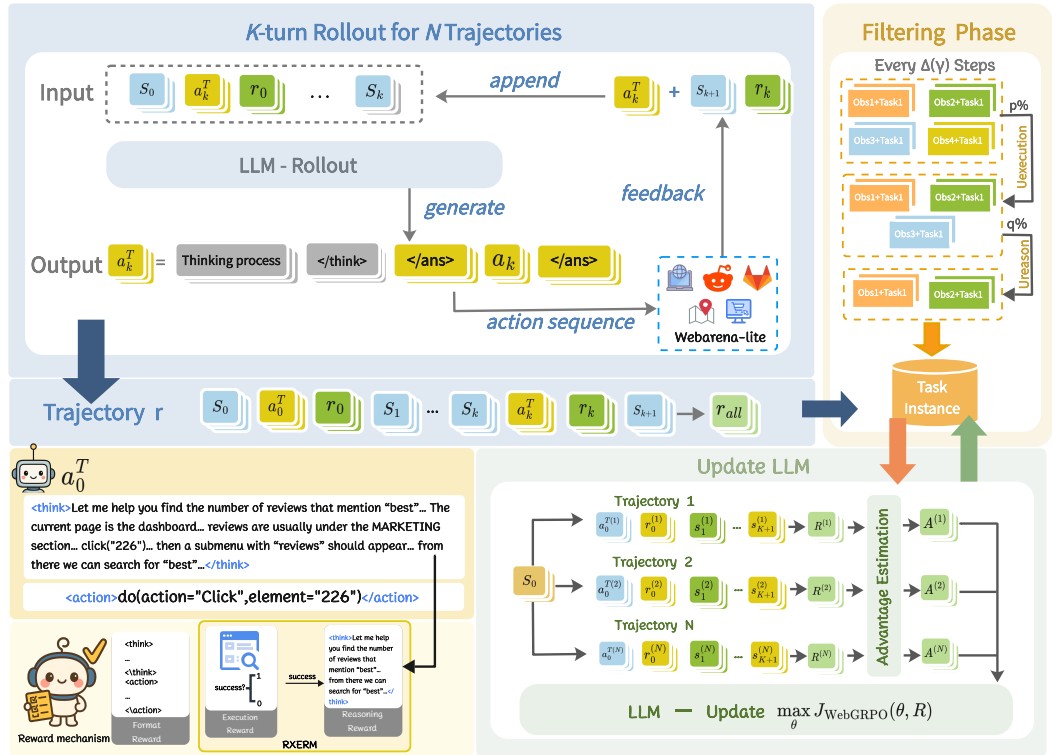

Figure 3: Overall WebGRPO pipeline, which leverages Two-Stage Reward mechanism RXERM and Dual-Uncertainty-Based Active Learning for Task Instance Filtering to achieve efficient training.

Where $\mathcal{M}$ is the finite-horizon Markov Decision Process (MDP), $\tau$ is a full sequence of reasoning-augmented interactions, and $R(\tau)$ is the total reward accumulated over trajectory $\tau$.

### 3.2.1 REASONING-INTERACTION TRAJECTORIES FOR WEB TASKS

In each training iteration, the agent begins with an initial state–task goal pair $(s_0, T)$, where a task instance corresponds to this specific pair, representing a concrete scenario for policy interaction and evaluation. Subsequently, the agent generates $N$ complete interaction trajectories and, at each time step $t$, produces a reasoning-guided structured output:

$$a_t^\top = \texttt{<think>...</think> <action>} \, a_t \, \texttt{</action>} \tag{4}$$

The `<think>` section records reasoning (e.g., element localization, form filling), and `<action>` specifies the executable web action $a_t$(e.g., `<Click [3]>`, `<Scroll Down>`). The web environment executes $a_t$, updates the state $s_{t+1}$, and returns an immediate reward $r_t$, forming a web interaction trajectory: $\tau = \{s_0, a_0^\top, r_0, s_1, a_1^\top, r_1, \ldots, s_K\}$. WebGRPO interleaves rollout and update steps. It employs an on-policy optimization approach, where trajectories are collected under the current policy $\pi_\theta$ for updates. In each training iteration, the agent samples from $P$ initial state–task goal pair $(s_0, T)$, collecting $N$ complete trajectories per state, resulting in $P \cdot N$ trajectories per iteration. Given a batch size of $E$, each iteration performs $\frac{P \cdot N}{E}$ gradient update steps. After $L$ training iterations, the total number of gradient updates is $S = L \cdot \left( \frac{P \cdot N}{E} \right)$.

### 3.2.2 TRAJECTORY-LEVEL GRPO OPTIMIZATION WITH RXERM

We propose a trajectory-level WebGRPO optimization framework based on Group Relative Policy Optimization (GRPO). Through format reward and a two-stage LLMs-based reward mechanism—RXERM (Reasoning & Execution Reward Model), rich and effective reward signals are provided for policy learning. A dual uncertainty-based active learning strategy further selects high-information task instances, improving the stability and performance of policy optimization.

**Token-Level Trajectory Definition.** In the WebGRPO framework, the policy probability $\pi_\theta(\tau)$ can be decomposed into token-level likelihoods due to the autoregressive nature of LLM-based agents. For every initial state–task goal pair $(s_0, T)$, multiple rollouts are sampled to collect complete interaction trajectories. The trajectory can be defined as: $\tau_i = \{\tau_{i,(1)}, \ldots, \tau_{i,(L)}\}$, where $\tau_{i,(t)}$ denotes the $t$-th token in trajectory $\tau_i$, and $L$ is the total number of tokens in the trajectory.

**Two-Stage Reward mechanism RXERM and Format Reward.** Under the same task and initial environment conditions, even if the agent successfully completes the task, many trajectories may still contain redundant or invalid actions. If only execution rewards are considered, it becomes difficult to effectively distinguish the quality among different successful trajectories. To address this limitation, we propose the two-stage LLMs-based reward mechanism—RXERM (Reasoning & Execution Reward Model) , which integrates both trajectory-level credit assignment and action-level credit assignment. RXERM separately evaluates both intermediate reasoning trace $\mathcal{R}$ (each <think> reasoning content) and task completion $\mathcal{E}$, providing fine-grained and comprehensive reward signals. Specifically, RXERM consists of two stages: In the first stage, Execution Reward Model evaluates task completion $\mathcal{E}$ based on the task goal $T$, trajectory $\tau$, and final state $s_K$, producing a binary execution reward $R_{\text{execution}}$ (1 for success, 0 otherwise). Only successful trajectories proceed to the second stage, where Reasoning Reward Model evaluates $\mathcal{R}$ using predefined prompts to compute intermediate reasoning reward $R_{\text{reason}}$, defined as the average reward over all reasoning steps along the trajectory. To separately assess action correctness and reasoning quality while ensuring output consistency, we introduce a widely used format reward. This encourages structured outputs using <think> for reasoning and <action> for actions. The final trajectory reward is computed as:

$$R(\tau) = \alpha R_{\text{reason}}(\tau) + (1 - \alpha) R_{\text{execution}}(\tau) + \beta R_{\text{format}}(\tau) \tag{5}$$

where $\alpha$ controls the balance between reasoning and execution rewards, and $\beta$ is a fixed weight for the format reward, used to provide basic structural constraints. This design ensures that the optimization process emphasizes high-quality reasoning–interaction trajectories rather than merely achieving task completion, thereby effectively preventing the agent from attaining non-robust success through repeated trials or accidental exploration.

**Dual-Uncertainty-Based Active Learning for Task Instance Filtering.** To improve policy optimization stability and efficiency, we propose a dual uncertainty-based active learning strategy for task instance filtering. Based on active learning theory (Settles, 2009), this strategy selects the most informative task instances by prioritizing those with high uncertainty, avoiding low-value trivial tasks and overly difficult tasks with weak reward signals. We jointly consider uncertainty in task completion $\mathcal{E}$ and intermediate reasoning trace $\mathcal{R}$ to guide task instance selection. Specifically, two uncertainty metrics are defined:

$$U_{\text{execution}} = \text{Std}_{\tau \sim \pi_\theta(\cdot|s_0, T)} \left[ R_{\text{execution}}(\tau) \right], \quad U_{\text{reason}} = \text{Std}_{\tau \sim \pi_\theta(\cdot|s_0, T)} \left[ R_{\text{reason}}(\tau) \right]. \tag{6}$$

Both uncertainties are computed as the standard deviation of rewards over multiple rollouts for the same task instance. In WebGRPO training, we first rank all task instances by $U_{\text{execution}}$ and retain the top $p\%$ with the highest uncertainty, where the corresponding threshold is denoted as $\delta_p$, i.e., $\mathcal{I}_{\text{execution}} = \{i \mid U_{\text{execution}}(i) \geq \delta_p\}$. From $\mathcal{I}_{\text{execution}}$, we further retain the top $q\%$ based on reasoning uncertainty $U_{\text{reason}}$, with threshold $\delta_q$, i.e., $\mathcal{I}_{\text{filtered}} = \{i \in \mathcal{I}_{\text{execution}} \mid U_{\text{reason}}(i) \geq \delta_q\}$. Only instances in $\mathcal{I}_{\text{filtered}}$ are used for policy optimization. Through this dual uncertainty filtering strategy, the model effectively eliminates low-information rollouts and significantly mitigates the risk of policy collapse in multi-turn reasoning scenarios. This filtering is not a one-time preprocessing step but is periodically repeated during training. We introduce a curriculum variable $\gamma$ to adaptively adjust the update interval $\Delta$. At each phase, after every $\Delta(\gamma)$ steps, uncertainty is recalculated and the task instances updated. With linear scheduling $\Delta(\gamma) = \Delta_0 + \gamma$, shorter intervals in early stages ensure frequent updates for fast learning, while longer intervals in later stages provide stable training. This balances efficiency within phases and curriculum-driven progression across phases.

**Trajectory Advantage Normalization.** Generate multiple trajectories for the retained high-quality instances, assign scalar rewards $R(\tau_i)$, and perform normalization within the batch. The normalized reward $\hat{A}_{i,t}$ is shared across all tokens within the same trajectory.

$$\hat{A}_{i,t} = \frac{R(\tau_i) - \text{mean}\left[ R(\tau_1), \ldots, R(\tau_G) \right]}{\text{std}\left[ R(\tau_1), \ldots, R(\tau_G) \right]}, \tag{7}$$

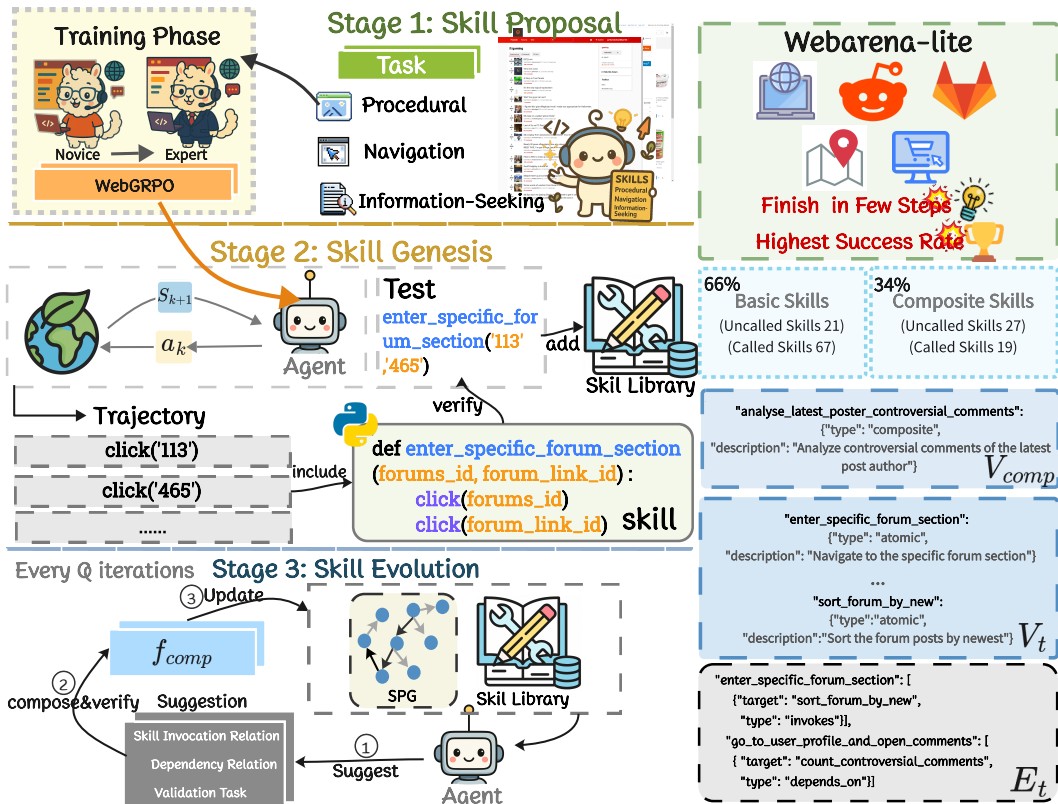

Figure 4: SkillGenesis, A Skill Evolution and Path Optimization Framework. WebGRPO can be seamlessly integrated into the SkillGenesis framework.

**Policy Optimization Objective.** The GRPO objective becomes:

$$J_{\text{GRPO}}(\theta) = \frac{1}{G}\sum_{i=1}^{G}\frac{1}{|\tau_i|}\sum_{t=1}^{|\tau_i|}\min\left[\rho_{i,t}\cdot\hat{A}_{i,t},\ \text{clip}(\rho_{i,t}, 1-\epsilon, 1+\epsilon)\cdot\hat{A}_{i,t}\right], \tag{8}$$

where $\rho_{i,t} = \frac{\pi_\theta(\tau_{i,(t)}|\tau_{i,<t})}{\pi_{\text{old}}(\tau_{i,(t)}|\tau_{i,<t})}$, G is the number of trajectories in the batch, $\tau_{i,(t)}$ denotes the $t$-th token in trajectory $\tau_i$, and $\tau_{i,<t}$ is its prefix.

### 3.3 SKILLGENESIS: DYNAMIC SKILL EVOLUTION WITH SKILL PATH GRAPHS

To enhance the agent's ability to complete tasks and improve skill reusability in complex web environments, we seamlessly integrate the WebGRPO into the SkillGenesis framework. This framework consists of three stages: Skill Proposal, Skill Genesis, and Skill Evolution, enabling dynamic expansion of the skill library through a periodic skill evolution mechanism and optimization strategies based on the Skill Path Graph (SPG), as illustrated in Figure 4(The figure is based on the case study in Appendix F) .

**Skill Library and SPG.** At time step $t$, the skill library $\mathcal{A}_t$ stores executable functions (Python programs) for low-level web actions (e.g., `enter_specific_forum_section()`), while the SPG $\mathcal{G}_t = (V_t, E_t)$ organizes skills as a directed graph, where $V_t$ represents nodes as structural abstractions (e.g., {"enter_specific_forum_section": type=atomic, description="Navigate to the specific forum section"}) and $E_t$ represents edges. Edges indicate either an invocation sequence or a dependency relation, such as "SearchItem $\mapsto$ Login" (dependency) and "SearchItem $\rightarrow$ AddToCart" (sequence). Dependency edges represent mandatory prerequisites, while sequence edges denote recommended but non-mandatory orders.

**Skill Proposal Stage.** In the Skill Proposal stage, LLM generates practical and reusable task goals $\{T_i\}_{i=1}^P$, based on $P$ initial webpage states $\{s_0^{(i)}\}_{i=1}^P$. These task goals cover common web interaction types, including procedural tasks, navigational tasks, and information-seeking tasks.

**Skill Genesis Stage.** In the Skill Genesis stage, the agent uses the WebGRPO policy to generate reasoning-interaction trajectories $\tau = \{s_0, a_0^\top, r_0, s_1, a_1^\top, r_1, \ldots, s_K\}$. When a task $T$ is successfully completed, we adopt a trajectory prefix rewriting method to induce skills. By truncating the original trajectory $\tau$, we construct a skill-invocation-based prefix $\tau_D = \{s_0, a_0, \ldots, s_D\}$. The agent continues to generate subsequent actions based on $\tau_D$, forming a complete and verifiable trajectory $\tau_f = \tau_D \cup \{a_{D+1}, \ldots, a_K\}$. The validity of the trajectory is evaluated by LLM based on three criteria: 1) *Correctness*: The task is successfully completed; 2) *Skill Usage*: New skills are utilized in the trajectory; 3) *Skill Effectiveness*: Skill invocations effectively change the environment state. If the criteria are met, the new cleaned skill set $\mathcal{D}_{\text{called}}$ is added to the skill library, updating: $\mathcal{A}_{t+1} = \mathcal{A}_t \cup \mathcal{D}_{\text{called}}$.

**Skill Evolution Stage.** The framework periodically triggers Skill Evolution stage after every fixed $Q$ SkillGenesis iterations. LLM proposes skill composition suggestions based on the current skill library $\mathcal{A}_t$ and the SPG $\mathcal{G}_t = (V_t, E_t)$. These suggestions include skill invocation relations, dependency relations, and corresponding validation tasks $T_{\text{comp}}$. Based on these suggestions, LLM utilizes the current SPG to composes candidate composite skills $f_{\text{comp}}$ and verifies them by executing $T_{\text{comp}}$. If the composite skill successfully completes the validation task, it is added to the skill library, updating

$$\mathcal{A}_{t+1} = \mathcal{A}_t \cup \{f_{\text{comp}}\}. \tag{9}$$

Simultaneously, the SPG is updated to reflect the new composite skill and its dependencies:

$$V_{t+1} = V_t \cup \{v_{\text{comp}}\}, \quad E_{t+1} = E_t \cup \{(v_i, v_j) \mid (v_i \to v_j) \text{ or } (v_i \mapsto v_j) \text{ discovered in } v_{\text{comp}}\} \tag{10}$$

where $v_{\text{comp}}$ denotes the new composite skill node. This periodic evolution mechanism ensures the skill library continuously accumulates high-quality composite skills, enhancing task efficiency and generalization capability. By continuously expanding and optimizing the skill library during exploration, the generated skills can be directly reused in subsequent tasks, effectively improving task performance in complex web environments.

# 4 EXPERIMENTS

## 4.1 ENVIRONMENTS AND BASELINES

### 4.1.1 ENVIRONMENT

To evaluate the performance of our proposed approach and baselines, we conducted experiments in the WebArena environment (Zhou et al., 2023). WebArena is an interactive benchmark designed for complex web navigation tasks, providing a self-hosted sandboxed web environment covering five major application domains: OpenStreetMap (Map), Reddit, GitLab, online store content management system (CMS), and OneStopShop (OSS). The full WebArena benchmark contains 812 instructions, but annotation errors and vague evaluation standards in the original dataset may undermine fairness and credibility. To ensure statistical reliability and lower the computational overhead of large-scale evaluations, we adopt WebArena-Lite (Lai et al., 2024), a human-verified subset of WebArena. This version selects 165 representative tasks as the evaluation set, providing a more consistent and trustworthy basis for assessment.

### 4.1.2 BASELINES

To evaluate the effectiveness of our method, we compare it against several baseline approaches, which can be divided into two categories: proprietary large language models utilizing prompting techniques, and open-source large language models trained with alternative paradigms. For fine-tuning methods, we adopt Llama3.1 (Grattafiori et al., 2024) and GLM-4-9B (GLM et al., 2024b) as backbone models. Following (Qi et al., 2025), the imitation learning (also known as supervised fine-tuning, SFT) baselines are trained on 9,460 trajectories from the human-labeled demonstration dataset of WebArena-Lite. For reinforcement learning baselines, we use the SFT-trained model as the initial actor, while the critic is also based on the SFT-trained model with an additional randomly

initialized value head. For the WebGRPO method, we employ a skill proposer during training to generate a large number of task instances, construct the initial task pool, and apply task instance filtering. After convergence, the policy $\pi_\theta$ is fixed, and the agent explores within the WebArena environment to continuously expand and validate the skill library, thereby adapting to more complex and dynamic scenarios. Specifically, during the SkillGenesis stage, the agent executes 160 exploration iterations, where newly discovered skills are incrementally added to the skill library. Every 20 iterations ($Q = 20$) of SkillGenesis, the Skill Evolution stage is triggered to evolve the skills using SPG and the existing skill library. The RXERM model used in reinforcement learning training and the LLM applied in the SkillGenesis stage both adopt GPT-4o. The rationale can be found in Appendix E.1 and Appendix E.6. More details of the training process can be found in Appendix B.

Table 1: Performance comparison of WebGRPO against baseline methods in terms of Task Success Rate (SR), evaluated on WebArena-Lite (Zhou et al., 2023; Liu et al., 2024). WebArena-Lite is a human-verified subset of the WebArena benchmark. (Results marked with * are reported from the literature on the full WebArena dataset.) The **best** and second-best models are indicated.

| Models | #Params | Reddit | Gitlab | CMS | Map | OSS | Avg. SR |
|---|---|---|---|---|---|---|---|
| *Proprietary LLMs* 🔒 | | | | | | | |
| GPT-4-Turbo (Achiam et al., 2023) | N/A | 10.5 | 16.7 | 14.3 | 36.7 | 13.3 | 17.6 |
| GPT-4o | N/A | 10.5 | 10.0 | 20.0 | 20.0 | 11.1 | 13.9 |
| AWM + GPT-4-0613* (Wang et al., 2024) | N/A | 50.9 | 31.8 | 29.1 | 43.3 | 30.8 | 35.5 |
| WebPilot + GPT-4o* (Zhang et al., 2025) | N/A | 65.1 | 39.4 | 24.7 | 33.9 | 36.9 | 37.2 |
| SkillWeaver+GPT-4o* (Zheng et al., 2025) | N/A | 50.0 | 22.2 | 25.8 | 33.9 | 27.2 | 29.8 |
| ASI+Claude-3.5-sonnet* (Wang et al., 2025) | N/A | 54.7 | 32.2 | 44.0 | 43.1 | 40.1 | 40.4 |
| *Open-sourced LLMs* 🔓 | | | | | | | |
| AutoWebGLM (Lai et al., 2024) | 6B | 9.4 | 15.0 | 28.6 | 24.8 | 17.1 | 18.2 |
| GLM-4-Chat (GLM et al., 2024a) | 9B | 5.3 | 10.0 | 6.7 | 3.3 | 6.7 | 6.1 |
| GLM-4 + SFT (BC) | 9B | 47.4 | 13.3 | 31.4 | 23.3 | 13.3 | 22.4 |
| GLM-4 + Filtered BC | 9B | 52.6 | 10.0 | 31.4 | 26.7 | 20.0 | 24.8 |
| GLM-4 + AWR (Peng et al., 2019b) | 9B | 52.6 | 16.7 | 34.3 | 30.0 | 22.2 | 27.9 |
| GLM-4 + DigiRL (Bai et al., 2024a) | 9B | 63.2 | 30.0 | 34.3 | 26.7 | 26.7 | 31.5 |
| GLM-4 + WEBRL (Qi et al., 2025) | 9B | 57.9 | 50.0 | 48.6 | 36.7 | 37.8 | 43.0 |
| GLM-4 + WebGRPO | 9B | 62.3 | 52.1 | 52.9 | 45.3 | 46.9 | 50.3 |
| GLM-4 + WebGRPO (+Skills) | 9B | 71.2 | **61.2** | 59.2 | **50.6** | 51.5 | 57.6 |
| Δ | | ↑14% | ↑18% | ↑12% | ↑12% | ↑10% | ↑15% |
| Llama3.1-Instruct (Grattafiori et al., 2024) | 8B | 0.0 | 3.3 | 2.9 | 3.3 | 11.1 | 4.8 |
| Llama3.1 + SFT (BC) | 8B | 36.8 | 6.7 | 20.0 | 33.3 | 17.8 | 20.6 |
| Llama3.1 + Filtered BC | 8B | 52.6 | 20.0 | 31.4 | 23.3 | 8.9 | 23.0 |
| Llama3.1 + AWR (Peng et al., 2019a) | 8B | 57.9 | 26.7 | 31.4 | 26.7 | 17.8 | 28.5 |
| Llama3.1 + DigiRL (Bai et al., 2024b) | 8B | 57.9 | 26.7 | 37.1 | 33.3 | 17.8 | 30.3 |
| Llama3.1 + WEBRL (Qi et al., 2025) | 8B | 63.2 | 46.7 | 54.3 | 36.7 | 31.1 | 42.4 |
| Llama3.1 + WebGRPO | 8B | 65.8 | 51.3 | 59.1 | 42.3 | 44.5 | 51.8 |
| Llama3.1 + WebGRPO (+Skills) | 8B | **75.8** | 60.2 | **65.3** | 48.1 | **53.9** | **60.4** |
| Δ | | ↑15.2% | ↑17.3% | ↑10.5% | ↑13.7% | ↑21.1% | ↑16.6% |

## 5 Main Results

Table 1 presents a comparison of WebGRPO with existing baseline methods on the WebArena-Lite benchmark. Our proposed method consistently outperforms all baselines across various environments, demonstrating the effectiveness of trajectory-level optimization and skill reuse. Specifically, for GLM-4, GLM-4 + WebGRPO (+Skills) achieves the highest average task success rate (SR) of 57.6%, surpassing the strongest baseline, GLM-4 + WEBRL (43.0%), by 14.6%, and outperforming all proprietary models. Notably, in the *Gitlab* and *Reddit* environments, compared to WebGRPO without skill reuse, the success rates further improve by +18.0% and +14.0%, respectively, demonstrating strong generalization and task adaptability. Similarly, for the Llama3.1 series, WebGRPO (+Skills) achieves an impressive average SR of 60.4%, outperforming the top baseline, Llama3.1 + WEBRL (42.4%), by 18.0%. In particular, the success rates in the *Gitlab* and *OSS* environments improve by +17.3% and +21.1%, respectively, compared to WebGRPO without skill reuse, further highlighting the significant advantages of skill reuse. Overall, these results demonstrate that WebGRPO, through the integration of reasoning-augmented interaction trajectories and efficient skill

Table 2: Quantitative ablation of the Skill Evolution Stage on GLM-4-9B and Llama-3.1-8B.

| Model | Config | Reddit | GitLab | CMS | Map | OSS | Avg |
|-------|--------|--------|--------|-----|-----|-----|-----|
| GLM-4-9B | No skills | 62.3 | 52.1 | 52.9 | 45.3 | 46.9 | 50.3 |
| | Base-only | 66.8 | 55.8 | 54.1 | 47.5 | 48.7 | 54.6 |
| | Base+Compose | 71.2 | 61.2 | 59.2 | 50.6 | 51.5 | 57.6 |
| Llama-3.1-8B | No skills | 65.8 | 51.3 | 59.1 | 42.3 | 44.5 | 51.8 |
| | Base-only | 69.7 | 55.2 | 61.3 | 44.1 | 49.4 | 55.9 |
| | Base+Compose | 75.8 | 60.2 | 65.3 | 48.1 | 53.9 | 60.4 |

library construction, significantly enhances the agent's reasoning capability and task completion performance in complex web environments.

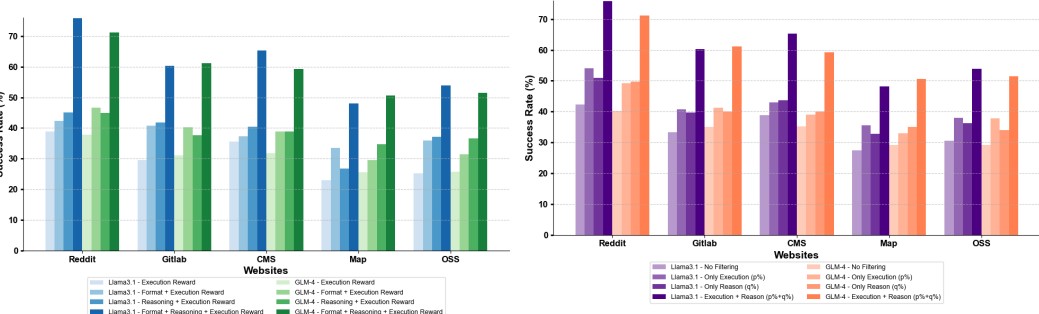

Figure 5: Quantitative Ablation Study on RX-ERM and Format Reward.

Figure 6: Quantitative Ablation Study on Dual-Uncertainty-Based Active Learning for Task Instance Filtering.

## 6 ABLATION STUDIES

**Reward Mechanism.** As shown in Figure 5, we performed ablation studies on Llama3.1 and GLM-4 to assess the two-stage RXERM reward mechanism and format reward. Removing the reasoning reward reduced average success rates by 7.62% and 7.42%, with the largest drop in *Reddit*, showing its importance for complex reasoning. Removing the format reward caused drops of 7.49% and 6.78%, confirming its role in stability. With only execution reward, rates fell to 30.42% and 30.36%, indicating that a single reward is insufficient.

**Dual-Uncertainty-Based Active Learning for Task Instance Filtering.** With Execution + Reasoning Reward Filtering, the success rates of Llama3.1 and GLM-4 increase to 60.4% and 57.6%, respectively, representing a substantial improvement over No Filtering. This demonstrates that the dual uncertainty strategy effectively boosts task success rate and stability (see Figure 6).

**Effect of Skill Evolution Stage.** To evaluate the effect of the skill evolution stage, we compared models under three configurations: *No skills*, *Base-only skills*, and *Base+Compose skills*. Results for GLM-4-9B and Llama-3.1-8B are shown in Table 2. Experimental results show that GLM-4-9B improved from 50.3% (no skills) to 54.6% (basic) and 57.6% (composite). Llama-3.1-8B similarly rose from 51.8% to 55.9% and 60.4%. Thus, the **Skill Evolution stage**, especially composite skills, is essential for improving performance on complex tasks. The detailed statistics of skill coverage, call frequency, and task-level skill call counts are provided in the Appendix D.

More detailed qualitative experiments can be found in Appendix E, including the evaluation of the Reward Mechanism RXERM in Section E.1, the analysis of model performance variance in Section E.2, the sensitivity analysis of the Reward Balancing Factor $\alpha$ in Section E.3, the sensitivity analysis of the Formatting Reward Weight $\beta$ in Section E.4, the sensitivity analysis of Dual-Uncertainty Task Filtering in Section E.5, and the performance comparison of SkillEvo with different LLMs in Section E.6. These sections provide detailed analysis and experimental results across various aspects of our method.

## 7 CONCLUSION

We propose **SkillEvo**, a framework combining trajectory-level optimization (WebGRPO) and dynamic skill evolution (SkillGenesis), which achieves state-of-the-art performance in long-horizon web tasks. Despite its effectiveness, our failure cases show limitations in dynamic environments

due to strong reward dependence, fragile skill composition, and high computational cost. Future work will focus on reducing training overhead, improving robustness of skill representations, and enabling continual learning with adaptive curricula for better scalability in real-world scenarios.

## ETHICS STATEMENT

This research strictly adheres to the ICLR Code of Ethics. No human subjects or animal experiments were involved, and therefore no ethical risks are present. All experiments were conducted on publicly available datasets (e.g., WebArena-Lite) in accordance with their usage agreements, and no personally identifiable information was used. In the design of our method and experiments, we carefully considered the avoidance of potential biases and discriminatory outcomes to ensure fairness and generalizability of the results across different models and environments. Furthermore, this study does not involve any activities that may cause privacy breaches, security risks, or inappropriate social impacts. We are committed to maintaining academic integrity and transparency throughout the entire research and publication process.

## REPRODUCIBILITY STATEMENT

To ensure the reproducibility of this work, we provide detailed descriptions of the experimental environment, dataset configurations, training procedures, and hyperparameter settings in the appendices (see Subsection4.1.1, Subsection4.1.2, Appendix B). All theoretical derivations, model details, and pseudocode are explicitly presented in the main text and appendices. In addition, upon acceptance, we will release the full source code and execution scripts on GitHub, enabling the research community to verify and reproduce our results. The released resources will include preprocessing scripts, implementations of WebGRPO and SkillGenesis, key hyperparameter configurations, and evaluation metric computation methods. We believe these resources provide sufficient detail for researchers to fully reproduce and extend our experimental results.

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

## A   THE USE OF LARGE LANGUAGE MODELS

During the preparation of this manuscript, we utilized a Large Language Model (LLM) as a writing assistant. The role of the LLM was strictly limited to language polishing, which included improving grammar, clarity, and overall readability. The LLM did not contribute to the research ideation, experimental design, methodology, or analysis of the results. All authors have reviewed the final text and take full responsibility for the content of this paper.

## B   TRAINING DETAILS

### B.1   DETAILS OF WEBGRPO

**Agent Actions.** The agent actions are mostly similar to those defined in *WebArena-Lite*(Liu et al., 2024):

- **Click**: Clicks an element with a specific ID.
- **Hover**: Hovers over an element with a specific ID.
- **Fill**: Types a message into an input box with a specific ID.
- **Search**: Types a message into an input box with a specific ID and presses Enter to initiate a search.
- **Keyboard_Press**: Emulates a specific keyboard key combination.
- **Scroll**: Scrolls the page up or down.
- **Select dropdown option**: Selects an option from a dropdown menu with a specific ID.
- **New tab**: Opens a new tab in the current browser.
- **Tab focus**: Switches focus to a browser tab at a specified index.
- **Close tab**: Closes the current tab.
- **Goto**: Navigates to a specific URL.
- **Go back**: Returns to the previous page.
- **Go forward**: Moves to the next page if available.
- **Exit**: Terminates the operation, returns the response, and exits.

**Detailed Training Process of WebGRPO.** During WebGRPO training with GLM-4 and Llama3.1 models, we configured 8×H100 GPUs and utilized the FSDP strategy. Each training starts with 8 state-task goal pairs ($P = 8$) and collects 16 trajectories ($N = 16$) per state. The dual uncertainty filtering is set to p=35% and q=45%, with $\beta = 0.4$ to balance reasoning and execution rewards. The maximum KL steps are limited to 10 ($K = 10$). In the periodic filtering schedule, we adopt a linear curriculum strategy with the initial update interval set to $\Delta_0 = 10$ and the curriculum index $\gamma \in 0, 10, 20, 30$, yielding update intervals $\Delta(\gamma) \in 10, 20, 30, 40$. The model trains for a total of 200 steps, optimizing the strategy for efficient and stable learning. The detailed hyperparameter settings are shown in Table 3. The detailed SkillGenesis framework is shown in Algorithm 1.

**Cost Analysis.** The training cost of the SkillEvo framework mainly comes from two stages: WebGRPO's trajectory-level optimization training and SkillGenesis's skill evolution exploration. In the training phase, WebGRPO used 8×H100 GPUs for approximately 38 hours of training, and SkillGenesis's exploration phase conducted 160 iterations, taking approximately 2.5 hours, with total training GPU hours of approximately 324 GPU hours. In terms of query costs, WebGRPO uses RXERM for trajectory reward modeling. For each successful trajectory, $1 + T$ GPT-4o calls are required (where $T$ is the number of reasoning steps, including $T$ reasoning evaluations and 1 execution evaluation); if the trajectory is unsuccessful, no reasoning evaluation will be performed. The SkillGenesis stage triggers one GPT-4o skill evolving call every 20 iterations. Based on the overall process estimation, the total call volume is approximately 101,000 GPT-4o calls. Among these, RXERM two-stage reward evaluation accumulates approximately 248M tokens (execution reward approximately 220M tokens, reasoning reward approximately 28M tokens). Additional overhead mainly comes from SkillGenesis bringing approximately 42M tokens (skill proposal one-time approximately 64k tokens, skill genesis approximately 42M tokens, skill evolving approximately 50k tokens).

---

**Algorithm 1** SkillGenesis Framework

---

1: **procedure** SKILLGENESIS($P, \{s_0^{(i)}\}_1^P$)
2:     $\{T_i\} \leftarrow$ `GPT4o.GenerateGoals`($\{s_0^{(i)}\}_1^P$)              ▷ Skill Proposal
3:     $\mathcal{A}_0 \leftarrow \emptyset; \mathcal{G}_0 \leftarrow (\emptyset, \emptyset)$          ▷ Initialize skill library and SPG
4:     **for** $t = 1$ to $T$ **do**
5:         $\tau \leftarrow$ `WebGRPO.GenerateTrajectory`($T$)         ▷ Skill Genesis
6:         **if** `Validate`($\tau, T$) **then**
7:             $\tau_D \leftarrow$ `TruncatePrefix`($\tau$)
8:             $\tau_f \leftarrow$ `ExtendWithPolicy`($\tau_D$)
9:             $\mathcal{D}_{\text{called}} \leftarrow$ `ExtractSkills`($\tau_f$)
10:           $\mathcal{A}_t \leftarrow \mathcal{A}_{t-1} \cup \mathcal{D}_{\text{called}}$          ▷ Update skill library
11:         **end if**
12:         **if** $t \bmod Q = 0$ **then**          ▷ Periodic Skill Evolution
13:             Suggestions $\leftarrow$ `GPT4o.ComposeSuggestions`($\mathcal{A}_t, \mathcal{G}_t$)
14:             **for** $f_{\text{comp}}$ in Suggestions **do**
15:                 **if** `Validate`($f_{\text{comp}}$.Task) **then**
16:                     $\mathcal{A}_t \leftarrow \mathcal{A}_t \cup \{f_{\text{comp}}\}$
17:                     $\mathcal{G}_t \leftarrow$ `UpdateSPG`($\mathcal{G}_t, f_{\text{comp}}$)
18:                 **end if**
19:             **end for**
20:         **end if**
21:     **end for**
22:     **return** $\mathcal{A}_t, \mathcal{G}_t$
23: **end procedure**

---

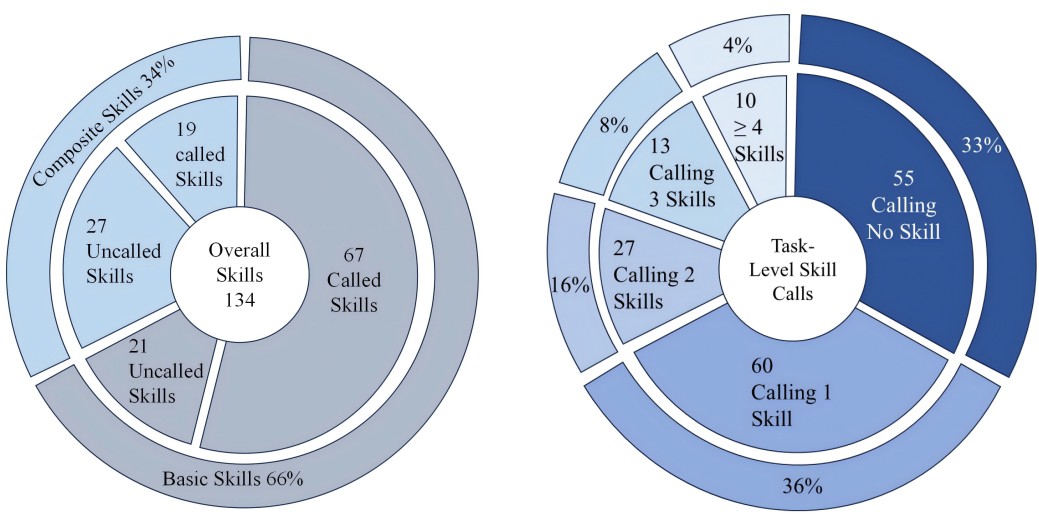

Figure 7: Overall statistics of skills, including distribution and usage frequency.

## B.2   HYPERPARAMETERS OF WEBGRPO AND BASELINES.

## C   SKILL ACCESS AND FILTERING

In this study, access to and filtering of the skill set is accomplished by directly listing the skill functions in the Prompt. Specifically, the 134 skill functions are divided into 5 batches, each containing about 27 functions, which are embedded in the `<functions>...</functions>` block of the Prompt. Based on the task description, the model determines whether each skill is relevant within every batch Prompt and outputs a JSON result containing step-by-step reasoning and function names. The results from the 5 batches are then merged and deduplicated to obtain a candidate skill

Table 3: The hyperparameters we employ in WEBGRPO and baselines.

| Method | Hyperparameter | Value |
|--------|----------------|-------|
| SFT | learning rate | 1e-5 |
| | lr scheduler type | cosine |
| | warmup ratio | 0.1 |
| | batch size | 128 |
| | training epoch | 1 |
| | cutoff length | 16384 |
| Filtered BC | learning rate | 1e-6 |
| | lr scheduler type | constant |
| | batch size | 128 |
| | training epoch | 1 |
| | cutoff length | 16384 |
| | filtering threshold | 70th percentile |
| AWR | actor learning rate | 1e-6 |
| | actor lr scheduler type | constant |
| | critic learning rate | 1e-6 |
| | critic lr scheduler type | constant |
| | batch size | 128 |
| | discount factor | 0.9 |
| | actor training epoch | 1 |
| | critic training epoch | 1 |
| DigiRL | actor learning rate | 1e-6 |
| | actor lr scheduler type | constant |
| | critic learning rate | 1e-6 |
| | critic lr scheduler type | constant |
| | instruction value function lr | 1e-6 |
| | instruction value function lr scheduler type | constant |
| | batch size | 128 |
| | discount factor | 0.9 |
| | actor training epoch | 1 |
| | critic training epoch | 1 |
| | instruction value function epoch | 1 |
| | rollout temperature | 1 |
| | replay buffer size | 100000 |
| WEBRL | actor learning rate | 1e-6 |
| | actor lr scheduler type | constant |
| | critic learning rate | 1e-6 |
| | critic lr scheduler type | constant |
| | batch size | 128 |
| | discount factor | 0.9 |
| | actor training epoch | 1 |
| | critic training epoch | 1 |
| | rollout temperature | 1 |
| WEBGRPO | actor learning rate | 1e-6 |
| | actor lr scheduler type | constant |
| | mini batch size | 64 |
| | discount factor | 0.95 |
| | rollout temperature | 1 |
| | validation temperature | 0.5 |
| | max prompt length | 4096 |
| | maximum response length | 512 |
| | KL-divergence loss coefficient | 0.001 |

set. A subsequent ranking Prompt is used to select the Top-15 most relevant skill functions, which are finally injected into the strategy model Prompt for task execution, as shown in Figure 20 and 21.

## D    Skills Distribution and Usage

As illustrated in Figure 7, in the complete 165 test tasks, we analyzed the coverage rate, call frequency, and complexity distribution of skill calls. After 160 exploration iterations, with composite skill generation triggered every 20 Genesis iterations, the final skill library size reached 134 skills, including 88 basic skills and 46 composite skills. In the test set, 86 skills were actually invoked, with 67 basic skills and 19 composite skills. The total number of skill calls was 193 (approximately 1.17 per task), with basic skill calls accounting for 141 (73%) and composite skills for 52 (27%). Although fewer in number, composite skills exhibited a higher reuse rate, with an average of 2.74 calls per skill, compared to 2.10 for basic skills. Task-level statistics show that 33% of tasks (55) can be completed entirely with atomic operations, without invoking any skills; 36% (60) invoked only 1 skill; tasks requiring 2 and 3 skills account for 16% (27) and 8% (13), respectively; high-complexity tasks requiring 4 skills constitute 6% (10). Thus, while simple tasks can often be solved with atomic operations or few basic skills, high-complexity tasks ($\geq 3$ skill calls) significantly depend on skill calls, contributing 41% of all invocations.

## E    Other Quantitative Experiments

### E.1    Evaluation of Reward Mechanismd RXERM

RXERM integrates **Execution Reward** (judging task completion) and **Reasoning Reward** (evaluating intermediate reasoning quality). We validated RXERM on 500 task trajectories using GPT-4o, comparing with human annotations and other evaluators (Claude-3.5-Sonnet, DeepSeek-R1, Gemini-1.5-Pro, GPT-4o-mini). As shown in Figure 8, GPT-4o achieved **92.7% execution accuracy** (second to DeepSeek-R1, 93.5%) and 92.1% reasoning accuracy (best, ~3% higher than Claude-3.5), leading to the highest overall accuracy (92.4%). Given its high consistency with human annotations, we ultimately chose GPT-4o as the RXERM model.

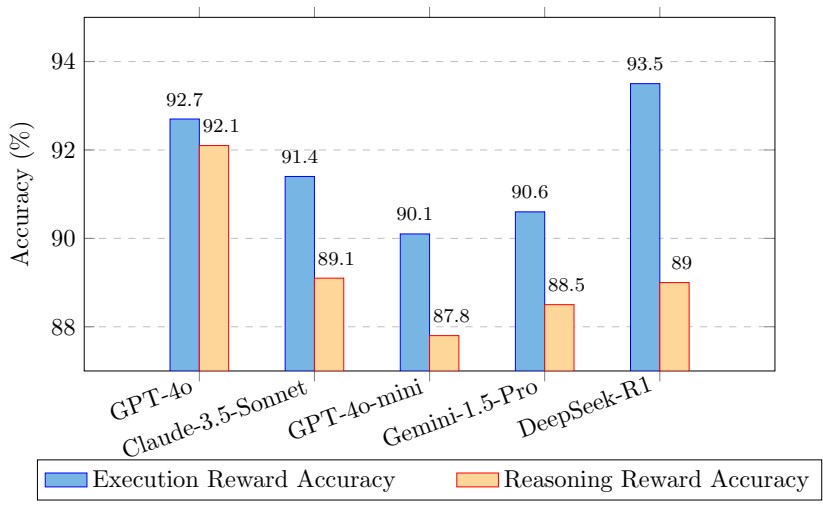

Figure 8: Execution Reward and Reasoning Reward Accuracy of different LLMs.

### E.2    Analysis of Model Performance Variance

Figure 9 presents the task success rates and error bars of various training methods on WebArena-Lite using the **GLM-4 (Left)** and **LLaMA3.1 (Right)** model series. Each method was evaluated with 5 runs using different random seeds to assess training stability and robustness. Due to the

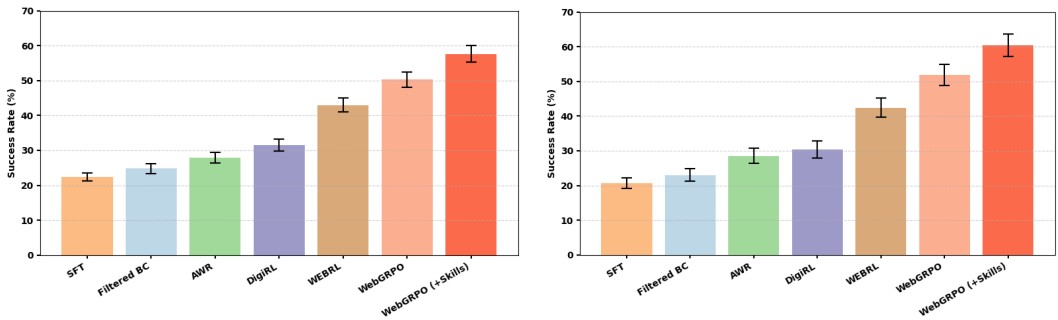

Figure 9: Task success rates and standard deviation error bars of different training methods on WebArena-Lite using the **GLM-4 (left)** and **LLaMA3.1 (right)** model series.

stochastic nature of Skill Proposal, WebGRPO exhibits relatively high variance. Nevertheless, both WebGRPO and its enhanced version, WebGRPO (+Skills), consistently outperform all baselines across both model series, achieving the highest average success rates. Even at the lower bound of the error bars, their performance remains superior, demonstrating the effectiveness and robustness of the proposed approach.

### E.3 SENSITIVE ANALYSIS REWARD BALANCING FACTOR $\alpha$

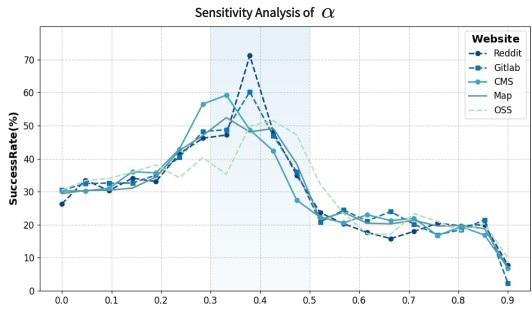

Figure 10: Sensitivity Analysis of $\alpha$

According to the sensitivity analysis, the choice of $\alpha$ has a significant impact on the success rate across different websites. As shown in Figure 10, the highest success rates for Reddit (0.38), Gitlab (0.38), CMS (0.34), Map (0.33), and OSS (0.42) are achieved when $\alpha$ is between 0.3 and 0.5, indicating the importance of increasing the weight of execution rewards. Both excessively low and high reasoning rewards reduce success rates—too low makes it hard to complete complex tasks, while too high may hinder the successful completion of trajectories. A proper balance between reasoning and execution rewards yields optimal performance.

### E.4 SENSITIVITY ANALYSIS OF FORMATTING REWARD WEIGHT $\beta$

In multi-turn tasks, reward signals are often sparse, delayed, and heavily outcome-based. As a result, models may produce the correct final answer through trial-and-error or shortcut strategies rather than coherent reasoning. We observe that in the absence of structural constraints, models frequently generate illogical or hallucinated reasoning traces—a phenomenon known as *reasoning collapse*. To address this, we introduce a formatting reward coefficient $\beta$, which penalizes outputs that do not conform to the expected `<think>`–`<answer>` structure, even when the final answer is correct. This encourages the model to maintain interpretable and logically consistent intermediate reasoning steps. As shown in Figure 11a, increasing the formatting reward weight to a moderate level (e.g., $\beta = 0.4$) leads to improved model stability and task success rate. Conversely, when

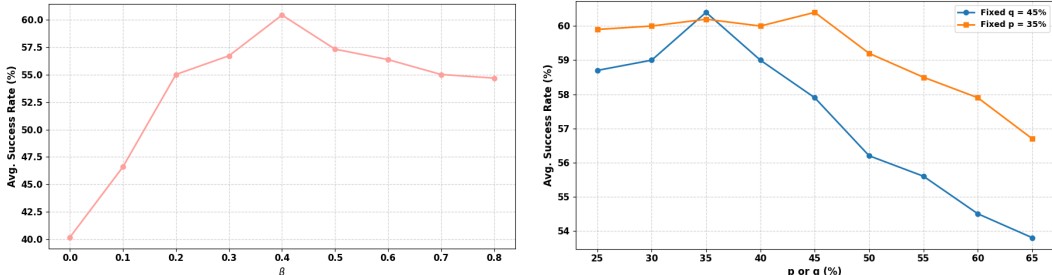

Figure 11: Sensitivity Analysis. **Left:** Sensitivity analysis of the formatting reward weight $\beta$. The plot shows the average task success rate across different values of $\beta$, which determines the influence of structural correctness in the overall reward. Performance improves significantly as $\beta$ increases, peaking at $\beta = 0.4$ with a success rate of 60.4%. A too-small or absent formatting reward leads to reasoning collapse and poor learning stability, while overly large values reduce performance due to excessive emphasis on structure. **Right:** Sensitivity analysis when varying percentile thresholds for execution and reasoning uncertainty. The figure shows the average task success rate under two settings: fixing the reasoning uncertainty percentile $q = 45\%$ (blue line) while varying $p$, and fixing the execution uncertainty percentile $p = 35\%$ (orange line) while varying $q$. The best performance (60.4%) is observed at $(p = 35\%, q = 45\%)$.

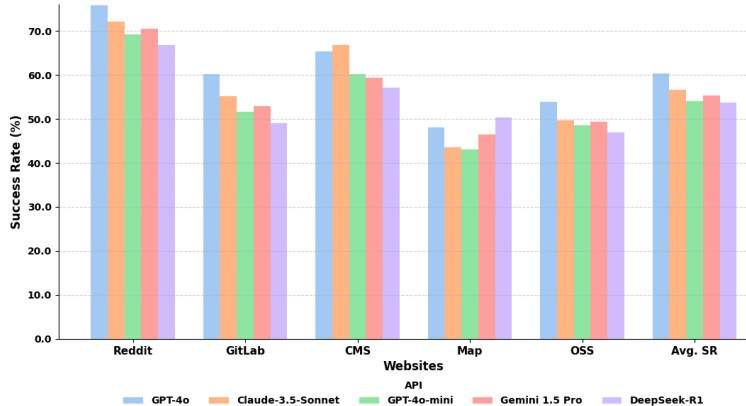

Figure 12: Success rate (%) of different LLMs on five web environments and the overall average in the SkillEvo workflow. GPT-4o achieves the highest average performance and leads on most tasks, while Claude-3.5-Sonnet slightly outperforms others on CMS and DeepSeek-R1 performs best on Map.

the formatting reward is too small or absent, the model's reasoning process often collapses, making policy learning unstable and more difficult to converge.

## E.5 SENSITIVITY ANALYSIS OF DUAL-UNCERTAINTY TASK FILTERING

We conduct a systematic sensitivity analysis over the execution uncertainty percentile $p$ and reasoning uncertainty percentile $q$. As shown in Figure 11b, the highest average task success rate (60.4%) is achieved when selecting the top $35\%$ of task instances based on execution uncertainty and further filtering the top $45\%$ based on reasoning uncertainty. This confirms that selecting moderately uncertain tasks contributes to both informativeness and stable policy optimization. In contrast, increasing either $p$ or $q$ leads to overly permissive filtering, allowing low-information or noisy instances to be included, which can destabilize training and even cause policy collapse. Conversely, excessively small $p/q$ values may discard too many useful samples, slightly lowering performance due to reduced diversity.

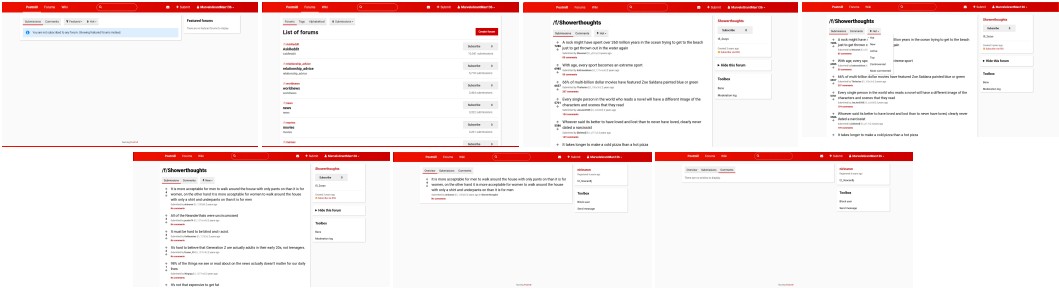

Figure 13: Reddit Example:Tell me the count of comments that have received more downvotes than upvotes for the user who made the latest post on the Showerthoughts forum.

### E.6   PERFORMANCE COMPARISON OF SKILLEVO WITH DIFFERENT LLMS

In LLaMA3.1 + WebGRPO (+Skills), we compare the performance of SkillEvo during the SkillGenesis stage when implemented with different large language models (LLMs). The figure presents the results of five LLMs—GPT-4o, Claude-3.5-Sonnet, GPT-4o-mini, Gemini-2.5-Pro, and DeepSeek-R1—across five web-based tasks (Reddit, GitLab, CMS, Map, OSS) and their average success rate (Avg. SR). The results show that GPT-4o consistently outperforms others on most tasks and achieves the highest average performance; Claude-3.5-Sonnet performs best on CMS, while DeepSeek-R1 slightly outperforms others on Map. Based on the overall performance, we ultimately select GPT-4o to implement the Skills component in SkillEvo.

## F   CASES STUDIES

To qualitatively analyze the impact of the SPG (Skill Path Graph) mechanism on action planning efficiency, we illustrate a representative Reddit example. Figure 13 presents the action trajectory for accomplishing the task "Tell me the count of comments that have received more downvotes than upvotes for the user who made the latest post on the Showerthoughts forum." . This multi-hop reasoning task requires understanding temporal posting order, user identity, and comment-level vote analysis. As shown in Figure 14, compared to the baseline agent (w/o SPG), which completes the task in four discrete steps, the agent with SPG (w/ SPG) achieves the same goal in a single step by optimizing the existing simple skills and composing them with more complex and advanced skills, which effectively improves the efficiency and cost of implementing web tasks.

In the initial stages of task execution, the agent identifies basic atomic skills such as `enter_specific_forum_section` (navigate to a specific forum section), `sort_forum_by_new` (sort posts by newest), and `go_to_user_profile_and_open_comments` (access user profile and open the comments section) to complete relatively simple tasks. As the Skill Path Graph (SPG) is gradually constructed, SkillEvo begins to recognize that these atomic skills are typically called in a specific order to achieve more complex goals. During the skill evolution process, these atomic skills are automatically integrated into more efficient composite skills. For example, the composite skill `analyse_latest_poster_controversial_comments` combines atomic skills such as `enter_specific_forum_section`, `sort_forum_by_new`, `go_to_user_profile_and_open_comments`, and `count_controversial_comments`, forming a complete execution workflow.

As shown in Figure15, the SPG clearly records the dependencies between skills (such as `depends_on`) and invocation edges (such as `invokes`), allowing the agent to replace the individual calls of 3 to 4 atomic skills with a single composite skill. This optimization significantly enhances the efficiency and stability of task execution. In this way, the SPG mechanism not only reduces the number of execution steps but also improves the flexibility and reliability of execution, thereby greatly improving the efficiency of web task execution and reducing costs.

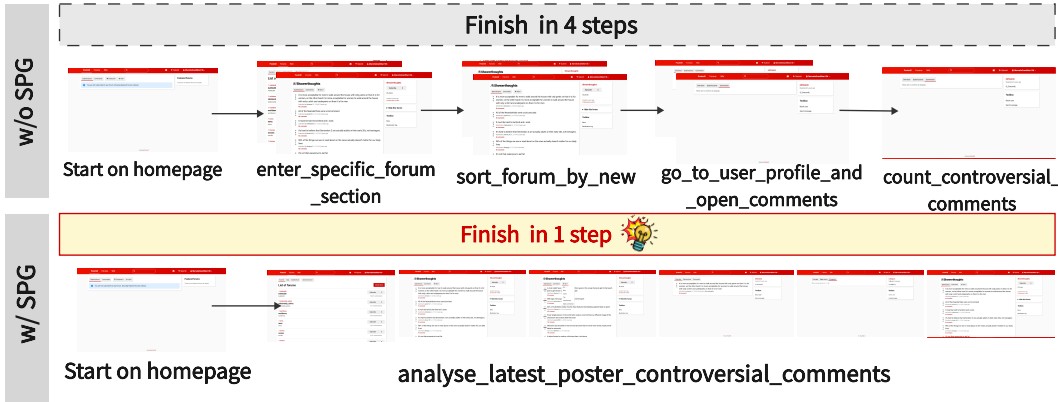

Figure 14: Workflow comparison between the w/o SPG and w/ SPG for task "Tell me the count of comments that have received more downvotes than upvotes for the user who made the latest post on the Showerthoughts forum.".

```
{
  "nodes": {
    "analyse_latest_poster_controversial_comments": {
      "type": "composite",
      "description": "Analyze controversial comments of the latest post author"
    },
    "enter_specific_forum_section": {
      "type": "atomic",
      "description": "Navigate to the specific forum section"
    },
    "sort_forum_by_new": {
      "type": "atomic",
      "description": "Sort the forum posts by newest"
    },
    "go_to_user_profile_and_open_comments": {
      "type": "atomic",
      "description": "Open the latest post author's profile and comments"
    },
    "count_controversial_comments": {
      "type": "atomic",
      "description": "Count comments with more downvotes than upvotes"
    }
  }
}
```

```
"edges": {
  "enter_specific_forum_section": [
    {
      "target": "sort_forum_by_new",
      "type": "invokes"
    }
  ],
  "go_to_user_profile_and_open_comments": [
    {
      "target": "count_controversial_comments",
      "type": "depends_on"
    }
  ]
}
```

Figure 15: Skill Path Graph (SPG) illustrating the nodes and edges for the task of "counting comments with more downvotes than upvotes for the user who made the latest post on the Showerthoughts forum". The nodes represent both atomic and composite skills, while the edges indicate the relationships between them, such as invocations and dependencies.

## G  FAILURE CASE ANALYSIS

Our preliminary analysis reveals two major categories of failure. First, **skill invocation degradation** is observed after WebGRPO training. Although the model can recognize appropriate skills during the exploration phase, in the execution phase it often reverts to atomic webpage operations instead of leveraging the skill library. For instance, in the task "view details of a book", the model should invoke the API `search_books_by_title('...')`, but instead fails to call it. This suggests that the policy lacks a global mechanism for evaluating the utility of skill calls, resulting in underutilization of evolved skills. Second, **parameter filling errors** are frequently encountered. In the task "find recipes containing broccoli but no dairy products", the model incorrectly generates the API call `search_recipes_by_ingredients('broccoli, milk-free')`, whereas the correct call should be `search_recipes_by_ingredients('broccoli', exclude='dairy')`. These errors typically stem from insufficient understanding of negation

and exclusion constraints, which leads to semantic misinterpretations when constructing API parameters. Together, these cases highlight current limitations of our approach, particularly in skill integration into policy execution and robust handling of fine-grained constraints.

## H DETAILS OF PROMPTS USED IN SKILLEVO

To enable the evaluation of reasoning and execution behaviors, as well as the generation and verification of skills for web-interacting agents, we have designed a series of carefully crafted prompt templates. These templates cover key aspects such as reasoning rewards, execution rewards, and the core components of the SkillGenesis framework, including skill proposal, validation, and composition.

As illustrated in Figure 22, we designed a reasoning quality evaluation prompt to assess the logical coherence, relevance, and interpretability of the agent's intermediate reasoning during task execution. This evaluation provides fine-grained reward signals. The execution success evaluation prompt, shown in Figure 23, determines whether the agent's sequence of actions successfully achieves the user's goal, serving as a key basis for execution rewards.

For skill generation, we introduced the SkillGenesis prompting system. The skill proposal prompt, shown in Figure 17, guides the agent to propose reusable and high-value interaction skills. The skill validation prompt, shown in Figure 16, evaluates the effectiveness and utility of proposed skills in specific tasks. Additionally, Figure 18 presents a prompt for abstracting reusable functions and rewriting action trajectories, promoting modularity and reusability in agent behaviors and Figure 19 shows a prompt for composing new skills from existing ones, supporting structured skill evolution and enhancing task efficiency. Together, these prompts form the core mechanism that supports the agent's learning and self-evolution process.

## I DEFINITION OF "SKILL"

A **skill** ($\sigma$) is defined as an executable Python function that takes two primary inputs: the current webpage state ($s$) and optional parameters ($\theta$), and returns an execution trajectory ($\tau$) consisting of multiple low-level actions such as clicking, typing, scrolling, etc. This can be expressed as the following formula:

$$\sigma : (s, \theta) \mapsto \tau \tag{11}$$

Where $s$ represents the current webpage state (or its abstract representation), $\theta$ represents optional parameters (such as product name, filter conditions, etc.), and $\tau$ is the execution trajectory composed of several low-level actions, such as clicks, typing, scrolling, etc.

Semantically, a **skill** represents a reusable high-level operation template that accomplishes specific tasks through a sequence of low-level actions. For example, a typical **skill** could be "search books by title," which is implemented in Python as follows:

```python
def search_books_by_title(title: str):
    # Includes several low-level actions.
    ...
```

During execution, the **skill** unfolds into a series of low-level actions, such as typing the book title into a search box and clicking the search button, thereby completing the task. In this way, the SkillGenesis framework abstracts complex web operations into high-level **skills**, enhancing task automation and reusability.

## LIMITATIONS AND FUTURE WORK

The current work demonstrates the effectiveness of Skill Path Graphs (SPG) in web environments, but there are several limitations that need to be addressed. Firstly, the experiments were primarily conducted in the WebArena-Lite environment, which includes five real-world websites, limiting the generalizability of the results to larger and more complex environments. While consistent improvements were observed across different websites and model families, future work will focus on

expanding the experiments to cover more websites and real-world scenarios to better assess the scalability and adaptability of the framework.

Secondly, although the core components of SkillEvo (such as WebGRPO + RXERM and SkillGenesis + SPG) are designed to be environment-agnostic, further validation is needed in embodied environments (e.g., long-range tasks based on Habitat/ALFRED) and mobile/GUI automation tasks (e.g., Android control or desktop automation) to evaluate the cross-domain generalization ability. This will help test the stability and effectiveness of the framework across different tasks and environments.

Additionally, as the number of skills increases, the pruning and subgraph retrieval strategies for SPG will face challenges. To handle large-scale skill sets, we plan to explore more efficient pruning algorithms and retrieval mechanisms to ensure that the system can effectively manage and optimize the skill path graph without compromising performance.

In summary, while the current system performs well in specific environments, future research will focus on improving its cross-environment generalization, expanding it to embodied and GUI environments, and optimizing the system to handle larger and more dynamic skill sets.

---

**Skill Verification Prompt:**

You are an expert in **verifying the validity of web interaction skills** generated by an autonomous agent. The agent attempts to complete a task using a combination of primitive actions and invoked skills. Please analyze the following based on the trajectory and invoked skills.

The user's goal is: {Task}

You will assess the trajectory against the following three criteria:

1. **Correctness** — Does the agent successfully complete the task goal?
2. **Skill Usage** — Does the trajectory include explicit invocations of reusable skills?
3. **Skill Effectiveness** — Do the invoked skills result in meaningful changes to the webpage state (e.g., navigation, DOM updates, successful form submission)?

## [Trajectory]: ## {State-action sequence; includes primitive actions and SKILL calls}

## [Invoked Skills]: ## {List of skills invoked in the trajectory}

For each criterion, respond with **YES** or **NO**, followed by a brief explanation. Finally, answer the following:

**Should the invoked skills be added to the skill library?**
Respond only with `YES` or `NO`.

---

Figure 16: Prompt for validating generated skills in the SkillGenesis framework.

---

**Skill Proposal Prompt:**

You are a **web agent** learning how to use a website. Your goal is to propose **"skills"** — Python functions that automate common tasks on the website. Each skill should act as a shortcut that combines multiple user interactions into a single reusable function. You are **not allowed** to interact with login/logout/account-related features on any site. If the site uses Magento, avoid the "Advanced Reporting" tab. If the site is OpenStreetMap, do not interact with community features.

You have already proposed the following skills:
<proposed>
##{procedural knowledge}##
</proposed>

The structure of the current website is provided below as HTML:
<html>
##{HTML structure}##
</html>

Please propose a new skill that reflects a task a real user might frequently perform. The skill should be meaningful and combine multiple interactions into a single callable function. **Avoid skills that perform only a single click.**

Follow these design guidelines for each skill:
- The skill name must be expressed in natural language.
- Do **not** use '*id' as a parameter.
- Each skill should simulate a multi-step sequence of interactions.
- **Prioritize tasks of the following types**:
    -Creating data (e.g., submitting a form)
    -Editing data (e.g., modifying entries)
    -Querying or filtering data (e.g., using search or filters)
- The total number of interactions (clicks, inputs, etc.) must **not exceed 10 steps**.

For each skill, evaluate it across the following three dimensions:
1. **Usefulness (1–3 points)**
    -3: A complex and frequently performed task with high user value.
    -2: A moderately frequent or moderately valuable task.
    -1: A rare or low-impact task.
2. **Generalizability (1–3 points)**
    -3: Can be reused across multiple pages or components.
    -2: Applies only to the current page but has a stable structure.
    -1: Depends on very specific or fragile HTML structure.
3. **Interaction Steps Score (1–10 points)**
    -Count the number of user interactions (click, input, select, etc.).
    -The more steps, the better — skills with more steps are more worthwhile to automate.
    -Maximum allowed steps: **10**

**Final Score = Usefulness + Generalizability + Number of Steps**

Your response must include a 'step by step reasoning' section explaining the three individual scores and listing each interaction step, followed by a 'proposed skill' section that names the highest-scoring skill using natural language.

Figure 17: Prompt used for skill proposer to propose reusable skills.

**Reusable Function Abstraction and Trajectory Rewriting Prompt:**

You are a proficient software engineer. Your task is to:
(1) **Summarize reusable functions as APIs** from the provided action trajectories;
(2) **Rewrite the trajectories using the reusable functions** you generated in (1).

**Step 1: Generate reusable functions**
From the provided trajectory, extract Python functions that encapsulate reusable and meaningful tasks.
- Each function should:
    - Contain **at least 3 actions**, but no more than 10 lines of code;
    - Be **general enough** for reuse in related scenarios;
    - Use **only common variables** as arguments (e.g., strings, lists), **not functions or complex objects**;
    - **Avoid try-except blocks**.
- The only valid actions are:

  • **Click**: Clicks an element with a specific ID.

  • **Hover**: Hovers over an element with a specific ID.

  • **Type**: Types a message into an input box with a specific ID.

  • **Search**: Types a message into an input box with a specific ID and presses Enter to initiate a search.

  • **Press**: Emulates a specific keyboard key combination.

  • **Scroll**: Scrolls the page up or down.

  • **Select dropdown option**: Selects an option from a dropdown menu with a specific ID.

  • **New tab**: Opens a new tab in the current browser.

  • **Tab focus**: Switches focus to a browser tab at a specified index.

  • **Close tab**: Closes the current tab.

  • **Goto**: Navigates to a specific URL.

  • **Go back**: Returns to the previous page.

  • **Go forward**: Moves to the next page if available.

  • **Exit**: Terminates the operation, returns the response, and exits.

- Each function must include the following in its docstring:
    - **Args** – Describe parameters;
    - **Returns** – Describe return values;
    - **Examples** – Show how the function is used.

**Step 2: Rewrite trajectories**
For each example:
- Provide an **instruction** describing the refactoring;
- Rewrite the original trajectory using the reusable functions;
- Do **not** include any response content or example-specific logic in the function calls.

**Important**:
- Make sure all used element IDs or URLs are visible in the original trajectory;
- If you use 'Exit', ensure the message is defined **within** the function;
- Each function should contain **2–10 steps only** to ensure simplicity;
- You may generate **zero, one, or multiple functions** depending on the input examples.

Figure 18: Prompt for reusable function abstraction and trajectory rewriting using WebArena-Lite action set.

---

**Skill Composition Prompt:**

You are an expert in skill modeling and composition. Your task is to help the system design a new composite skill by combining several existing skills in the skill library. The goal is to improve the efficiency and reusability of web agents in completing complex tasks.

Current Skill Library:
##{skill1, skill2, skill3, ...}##

##Skill Path Graph (SPG):##
- List of skill nodes:
<vt> ##{vt}## </vt>
- List of skill dependency edges:
<et> ##{et}## </et>

In the SPG:
$\rightarrow$ means sequential invocation, e.g., `f1` $\rightarrow$ `f2` $\rightarrow$ `f3`;
$\rightsquigarrow$ means strong dependency, e.g., `f2` $\rightsquigarrow$ `f3`.

**Your Task:**
1. Select **2–4 existing skills** and design a new composite skill $f_{comp}$;
2. Assign a **natural language name** to the skill;
3. Explain:
- What this skill does;
- When it is useful;
- Why this combination makes sense;
4. Specify:
- The invocation order: `f1` $\rightarrow$ `f2` $\rightarrow$ `f3`;
- Any strong dependencies: `f2` $\rightsquigarrow$ `f3`;
- A validation task $T_{comp}$ that can test this skill.

**Output Format:**
1.**Skill Name**: <natural language name>
2.**Composed From**: <list of component skills>
3.**Invocation Order**: <e.g., f1 $\rightarrow$ f2$\rightarrow$ f3>
4.**Dependency Edges**: <e.g., f2$\rightsquigarrow$ f3> (optional)
5.**Validation Task**: <task to test the skill>
6.**Why Useful**: <brief explanation>

**Notes:**
1.The composite skill must have **no more than 10 steps**;
2.Only propose meaningful, reusable, and testable combinations;
3.Return nothing if no reasonable skill composition exists;
4.Focus on complete user flows, frequent patterns, or performance benefits;
5.The skill must be verifiable by outcome.

Figure 19: Prompt for proposing composite skills in the Skill Evolution stage.

**Skill Filtering Prompt:**

You are provided with a list of Python functions that represent action shortcuts available on a website (in addition to basic actions like click, type, hover, select option, etc.).

##Function Space:##
<functions>...</functions>

##Task Description:##
{repr task}

**Your Task:**
1. Analyze each function and determine whether it is relevant or potentially useful for the task (not only the most obvious ones).
2. Include all functions that could be useful, even if they are not strictly necessary but might help in completing the task.
3. Output the result in the following JSON format:
{ "step by step reasoning": "Explain your reasoning process", "function names": ["function_name_1", "function_name_2", "function_name_3", ...] }

Figure 20: Prompt for filtering skills in the SkillGenesis framework.

**Skill Ranking Prompt:**

Now you have a list of candidate functions:
{candidate functions list}

##Task Description:##
{repr task}

**Your Task:**
1. Rank these functions based on their relevance to the task;
2. Return the Top-15 most relevant functions;
3. Output the result in the following JSON format:
{ "step by step reasoning": "Explain how you determined the ranking", "function names": ["Top1_function_name", "Top2_function_name", ..., "Top15_function_name"] }

**Notes:**
1. Ensure reasoning is clear and directly tied to task requirements;
2. Return only valid JSON structures;
3. If fewer than 15 functions are relevant, return all that apply.

Figure 21: Prompt for ranking skills in the SkillGenesis framework.

**Reasoning Quality Evaluation Prompt:**

You are an **expert reasoning evaluator for web-based interactive agents**. Your task is to analyze the agent's intermediate reasoning and assess its quality for reward feedback. Follow the evaluation criteria carefully.

##[Task Goal]:## {Task}
##[Current Webpage State]:## {Html of Current State}
##[Interaction History]:## {Action History}
##[Agent's Reasoning Trace]:## {Reasoning Trace}

Please evaluate the reasoning based on the following four criteria, assigning a score between 0 and 1 for each:
1. **Relevance to Task Goal (0–1)**
- Does the reasoning clearly connect to the specified task goal?
- Are the identified webpage elements or proposed reasoning directly helpful for achieving the task?
2. **Understanding of Webpage State (0–1)**
- Does the reasoning accurately describe relevant elements on the current webpage (e.g., buttons, forms, navigation bars)?
- Is there clear evidence that the agent understands the current structure and state of the webpage?
3. **Logical Consistency of Reasoning (0–1)**
- Does the reasoning exhibit a clear and coherent causal relationship between observations and planned actions?
- Are the proposed actions logically justified, explaining why these actions are necessary to achieve the task goal?
4. **Interpretability and Intermediate Steps (0–1)**
- Does the reasoning explicitly describe intermediate goals or sub-tasks rather than jumping directly to the final goal?
- Are the intermediate steps clearly explained, with reasoning about why each step is necessary?

##[Final Reasoning Quality Score] (Weighted average of the four criteria, range [0, 1]):## {Final Score}

##[Brief Explanation of Evaluation]## (Why did you assign this score?):

**IMPORTANT:** Only provide the final numerical score and a concise explanation. Do **not** solve the task direct.

Figure 22: Prompt for evaluating agent reasoning quality.

**Task Completion Evaluation Prompt:**

You are an expert in **evaluating the performance of a website navigation agent**.
The agent is designed to help a human user navigate the website to complete a task.
Please observe the following action history of an agent assisting a user on a website.
The user's goal is: {Task}
Based on the agent's action history and the final screen state, your goal is to determine whether the agent successfully completed the task.
Respond only with YES or NO.
##[Interaction History]:##
{Action History}
##[Final Webpage State]:##
{Html of Final Webpage State}

Figure 23: Prompt for evaluating agent's task completion status.

