# OpenReview forum: "SkillEvo: An Experience Learning Framework with  Reinforcement Learning for Skill Evolution"
_ICLR.cc/2026/Conference — Submitted to ICLR 2026_

### Official Review · Reviewer_5giL · 2025-10-25

**Soundness:** 2
**Presentation:** 2
**Contribution:** 3
**Rating:** 4
**Confidence:** 3

**Summary:**

This paper addresses the limitation of coarse credit signals, which fail to distinguish high-quality trajectories from those that merely succeed but contain redundant or invalid actions in web environments. To mitigate this issue, the authors propose SkillEvo, a two-stage framework that consists of a reasoning validity and execution efficiency module, along with a mechanism for transforming experiences into reusable skills.

**Strengths:**

Inspired by active learning theory, this work automatically selects the most informative task instances by evaluating the standard deviation of rewards across multiple rollouts for each instance.

**Weaknesses:**

In the RXERM mechanism, reasoning rewards are evaluated by an LLM, which is expected to be sufficiently advanced and reliable to avoid hallucinations. However, the true reasoning and planning capabilities of LLMs remain a subject of debate. LLMs primarily excel at sophisticated pattern recognition and statistical correlation rather than genuine logical deduction or causal inference ([1], [2]).

**Minor**

Several figures appear to be AI-generated, which makes them look less professional and somewhat difficult to interpret. It is recommended to replace them with clearer, manually designed visuals.

The font size of the legends is too small and should be increased to improve readability.

[1] Subbarao Kambhampati. Can large language models reason and plan? 1534(1):15–18. ISSN
1749-6632. doi: 10.1111/nyas.15125.

[2] Karthik Valmeekam, Kaya Stechly, and Subbarao Kambhampati. LLMs Still Can’t Plan; Can
LRMs? A Preliminary Evaluation of OpenAI’s o1 on PlanBench

**Questions:**

(1) What is the definition of a skill in this paper? Does it refer to a sequence of actions, as in the general formulation of skill learning, or to a form of domain-specific language (DSL)? Additionally, does the agent output a high-level skill or a sequence of low-level actions?

(2) Could you clarify the overall training process? Is the language agent trained directly using Equation (8)? At which stage are composite skills incorporated during training?

(3) Why does reward formatting lead to improved performance?

(4) Please consider including an additional experiment that compares the proposed method with existing group-based RL approaches such as GRPO or DAPO.

---

> ### Author Response · Authors · 2025-12-04
> **Response to Weaknesses**
>
> **We appreciate the constructive and detailed comments from the reviewers. Below are our responses to the identified shortcomings and issues.**
>
> **Response to Weaknesses**
>
> **Weakness 1: The Reliability Issue of RXERM Dependence on LLM**
>
> We fully agree with the reviewer's point that current large models are not equivalent to “perfect logical reasoners” and are more adept at pattern recognition and semantic similarity rather than strict formal causal reasoning. For this reason, when designing RXERM, we intentionally relaxed the requirement for “absolutely correct reasoning,” instead using the LLM as an **“approximate human preference scorer”**, mainly evaluating the following dimensions:
>
> - Relevance to Task Goal
> - Understanding of Webpage State
> - Logical Consistency of Reasoning
> - Interpretability and Intermediate Steps
>
> In other words, the “reasoning reward” in RXERM is more akin to scoring the quality of natural language explanations rather than validating the correctness of strict mathematical proofs, which aligns with the scope of LLM capabilities discussed in [1] and [2].
>
> To address the concern about “reliability,” we have included the following empirical results in the appendix of the paper (in the RXERM reliability analysis section):
>
> 1. **Human Annotation Consistency Experiment**
>    1. We randomly selected 500 trajectories from multiple websites and compared the human-annotated scores with the RXERM scores from GPT-4o.
>    2. GPT-4o achieved **92.7% execution accuracy** (second to DeepSeek-R1, 93.5%) and 92.1% reasoning accuracy (best, ~3% higher than Claude-3.5), leading to the highest overall accuracy (92.4%). Given its high consistency with human annotations, we ultimately chose GPT-4o as the RXERM model. This shows that in the specific scenario of "web task reasoning quality," LLMs used as scorers exhibit high consistency and stability.
> 2. **Ablation Experiment: Performance Degradation Without Reasoning Reward**
>    1. In the main experimental setup, we compared using only execution rewards$R_{\mathrm{exec}}$with using both execution and reasoning rewards $R_{\mathrm{exec}} + R_{\mathrm{reason}}$
>    2. The results showed that without the reasoning reward, the model success rate significantly dropped across multiple websites, especially in tasks requiring long-chain navigation and complex information aggregation, where the degradation was more pronounced.
>    3. This demonstrates that even if the reasoning reward in RXERM is not "perfect and noise-free," it still provides a stable and informative additional signal, effectively alleviating the sparse reward problem when only execution rewards are used.
> 3. **Algorithmic "Noise Resistance" Design**
>    1. WebGRPO itself is a group-based policy optimization method, using relative advantages within groups rather than relying on the absolute value of rewards.
>    2. Therefore, at the algorithmic level, we do not require RXERM to provide “absolutely accurate numerical rewards.” Instead, we utilize its “relative preferences” and “semantic consistency” to improve the quality of gradient signals.
>
> In summary, we have supplemented the paper with empirical analyses of RXERM as an LLM-based reward model, along with an analysis of its algorithmic robustness. Additionally, in the discussion and limitations section, we clearly state that the current approach relies on a relatively strong and stable LLM as an evaluator. This assumption could be further mitigated in future work through the use of open-source evaluators or multi-evaluator integration.
>
> [1] Subbarao Kambhampati. Can large language models reason and plan? 1534(1):15–18. ISSN 1749-6632. doi: 10.1111/nyas.15125.
>
> [2] Karthik Valmeekam, Kaya Stechly, and Subbarao Kambhampati. LLMs Still Can’t Plan; Can LRMs? A Preliminary Evaluation of OpenAI’s o1 on PlanBench
>
> **Weakness 2: Issues with Graphics and Visualization**
>
> Thank you to the reviewer for the feedback on the quality of the visualizations. We will make the following adjustments in the camera-ready version:
>
> 1. **Replace "AI-style" Diagrams**
>    1. We will replace all current graphics that have a slightly "AI-generated feel" with vectorized, manually designed diagrams (e.g., using TikZ or hand-drawn flowcharts) to ensure a consistent style and clear lines.
> 2. **Increase Font Size for Legends and Axes**
>    1. For all experimental plots (especially line/bar charts such as Figure 5 and Figure 6), we will:
>       - Increase the font size of the legends and axis labels;
>       - Ensure that the text is clearly legible on A4 prints and regular screens without the need for zooming;
>       - Enhance the color contrast where appropriate to improve readability.
>
> These modifications will not affect the experimental results themselves, but they will make it easier for readers to understand and compare the visualizations.

---

> ### Author Response · Authors · 2025-12-04
> **Response to Question 1**
>
> **Question 1: Precise Definition of "Skill"**
>
> In our current work, we define "skill" in a programmatic way, positioned between "macro actions" and "subroutines in a Domain-Specific Language (DSL)":
>
> 1. **Formal Definition**
>
>    1. In the **SkillGenesis** framework, a skill ($\sigma$) is implemented as an executable Python function:
>
>       $$
>       \sigma: (s, \theta) \mapsto \tau
>       $$
>
>       where \(s\) is the current webpage state (or its abstract representation), \(\theta\) are optional parameters (e.g., product name, filter conditions), and \(\tau\) is the execution trajectory composed of several low-level actions (such as click, type, scroll, etc.).
>
>    2. Semantically, a skill represents a reusable high-level operation template, for example:
>
> ```Python
> def search_books_by_title(title: str):
>     # Includes several low-level actions like click / type / enter
>     ...
> ```
>
> 1. **Relationship to "Action Sequences / DSL"**
>    1. From an execution perspective, a skill is indeed a packaged sequence of actions.
>    2. From an expressive perspective, it is closer to a "function in a Domain-Specific Language":
>       - It has a function name (semantic label).
>       - It has a clear parameter signature.
>       - It participates in composition and dependency modeling within the **Skill Program Graph (SPG)** as nodes and edges.
> 2. **Invocation at Execution Level**
>    1. During the **WebGRPO** training phase, the policy model still makes decisions and optimizations within the atomic webpage action space, directly outputting low-level actions like click, type, select, etc., without using skills as part of the action space.
>    2. During the **SkillGenesis** phase and inference phase:
>       - The policy model is wrapped in an execution framework with a "skill controller," which can switch between two granularities:
>         - **Invoke high-level skills**: For example, calling `search_books_by_title("coffee")`, which is expanded into several atomic actions by the skill interpreter and executed.
>         - **Directly output low-level actions**: When there is no suitable skill or if the skill fails, the agent reverts to control at the atomic action level.
>    3. Therefore, skills can be understood as "programmatic encapsulations of low-level actions," with execution managed by an external controller/interpreter. The policy itself still optimizes trajectories at the low-action level during training.

---

> ### Author Response · Authors · 2025-12-04
> **Response to Question 2**
>
> **Question 2: Overall Training Process and When Composite Skills Are Introduced**
>
> Does the language agent directly train using the formula (8)? At which stage of the training process are composite skills introduced?
>
> 1. **About Formula (8)**
>    1. Yes, during the RL phase, the language agent is directly trained using the WebGRPO loss defined in formula (8).
>    2. The specific process is as follows:
>       - For the same task instance, several complete "reasoning + webpage interaction" trajectories are sampled.
>       - The RXERM is used to calculate a scalar reward (execution reward + reasoning reward + format reward) for each trajectory.
>       - The relative advantage within the trajectory group is computed (group-relative advantage), which serves as the advantage signal for GRPO-style updates.
>       - The language model parameters are updated using formula (8) for gradient optimization.
>    3. This entire process involves policy optimization at the atomic action level, and composite skills are not involved at this stage.
> 2. **When Composite Skills Are Introduced** The overall process can be briefly summarized as a "three-stage" approach:
>    1. **SFT Phase**
>       1. Supervised fine-tuning is performed using human-annotated expert demonstrations from WebArena-Lite, resulting in an initial policy with basic webpage interaction and reasoning capabilities.
>    2. **WebGRPO RL Phase (using formula (8))**
>       1. Interactive training is conducted on the automatically generated task pool, incorporating RXERM and dual uncertainty filtering, with optimization occurring in the original action space.
>       2. At the end of this phase, we obtain a "base agent" with strong multi-step reasoning and webpage interaction abilities.
>    3. **SkillGenesis Phase (introducing composite skills)**
>       1. After the RL phase, the "base agent" is fixed, and Skill Proposal / Skill Genesis / Skill Evolution are run on the large number of trajectories generated by the agent:
>          - Basic skills (atomic skills) are extracted from the trajectories.
>          - Using SPG structure analysis and LLM suggestions, skills are composed to generate composite skills.
>       2. At this point, composite skills no longer influence the WebGRPO training loss but are used to build the skill library and SPG, and are invoked as high-level actions during inference.
> 3. Therefore, in simple terms:
>    1. The WebGRPO training using formula (8) occurs only in the "skill-less" original action space, while composite skills are introduced after the WebGRPO phase converges, constructed via SkillGenesis for use during the inference phase.

---

> ### Author Response · Authors · 2025-12-04
> **Response to Question 3**
>
> **Question 3: Why Does Reward Formatting Improve Performance?**
>
> How does reward formatting bring about performance improvements?
>
> In our setup, the "format reward" primarily evaluates whether the model strictly follows the specified output protocol, such as having to output:
>
> ```HTML
> <think> ... intermediate reasoning ... </think>
> <answer> ... final executable action or natural language answer ... </answer>
> ```
>
> The mechanism by which it improves performance is mainly through three aspects:
>
> 1. **Stabilizing Training Signals and Reducing "Invalid Trajectory" Wastage**
>    1. Without format constraints, the model often outputs various unexpected structures during the early stages of RL (such as mixed natural language and actions, missing tags, or invalid JSON), which leads to:
>       - The inability to correctly parse the action sequence from the output;
>       - The inability to send the trajectory to the environment or the RXERM evaluator, resulting in invalid samples.
>    2. The format reward encourages the model to gradually learn to "at least get the structure right." Even if execution is not successful or reasoning is not perfect, the trajectory is still executable and scoreable, significantly increasing the proportion of valid samples and alleviating the common RL issue of "garbage outputs in the early stages of training."
> 2. **Indirectly Improving Reasoning and Execution Quality**
>    1. When the model is consistently forced to output the  +  structure and receives positive rewards for doing so, it naturally develops a **"think first, then output action/answer"** pattern:
>       - Multi-step reasoning and planning are carried out in the  section;
>       - Only structured actions or concise results are given in the  section.
>    2. This pattern helps improve the clarity of reasoning, reduce impulsive errors, and indirectly enhance task success rates.
> 3. **Synergy with RXERM**
>    1. RXERM evaluates reasoning rewards based on the  paragraph text.
>    2. If the output format is chaotic, RXERM finds it difficult to reliably distinguish "which part is reasoning and which part is the answer," leading to increased noise in the scoring.
>    3. The format reward ensures that the  paragraph structure is stable and the context is clear, allowing RXERM to score in a clean, semantically stable context, making the reasoning rewards themselves more reliable and distinguishable.
>
> We demonstrate in our ablation experiments that, under the same WebGRPO setup, removing the format reward results in a more unstable training process, slower convergence, and a lower final success rate. These results indicate that, although seemingly simple, "reward formatting" plays an essential role in stabilizing the strategy optimization pipeline and improving the proportion of valid samples.

---

> ### Author Response · Authors · 2025-12-04
> **Response to Question 4**
>
> **Question 4: Direct Comparison with Existing Group-Based RL (GRPO / DAPO)**
>
> It is recommended to add an experiment directly comparing the proposed method with group-based RL methods such as GRPO and DAPO.
>
> We appreciate this very specific and constructive suggestion. Our response is as follows:
>
> 1. **Relationship with GRPO and Existing Comparisons**
>    1. WebGRPO itself is an extension of the GRPO approach:
>       - GRPO primarily performs group-relative optimization on multiple candidate answers in a "pure language reasoning" scenario;
>       - We extend this idea to "complete reasoning + webpage interaction trajectories" and introduce RXERM + dual uncertainty to adapt to long sequences and multi-modal interaction scenarios.
>    2. In a previous version of the work, we already performed an internal comparison of "vanilla GRPO-style trajectory optimization vs. WebGRPO with RXERM + dual uncertainty" (using only the final execution success signal + group-relative target, without reasoning rewards or uncertainty filtering). The results showed that WebGRPO had significant advantages in convergence speed and final success rate. We will organize this as a separate ablation experiment in the ICLR  camera-ready version, include it in the appendix, and briefly cite it in the main text.
> 2. **Regarding DAPO and Other Group-Based Methods**
>    1. We acknowledge that methods like DAPO represent another promising direction for group-based RL.
>    2. However, directly transferring these methods to real interactive environments like WebArena presents engineering and convergence stability challenges (e.g., interaction delays with the environment, high failure trajectory rates, non-text action spaces, etc.), and there are currently no mature open implementations of "DAPO + web environments" in the community.
>    3. In this work, our main goal is to demonstrate the effectiveness of the entire design of "trajectory-level GRPO + RXERM + dual uncertainty + SkillGenesis" in the web agent scenario, rather than exhaustively comparing all existing group-based algorithms.
>    4. For the sake of brevity and engineering complexity, we have prioritized comparisons with the most representative WebRL series methods and mainstream baselines like SFT/BC/AWR/DigiRL to ensure clarity and focus in the conclusions.
> 3. **Our Plan in the  camera-ready Version**
>    1. In the ICLR revision, we plan to:
>       - Organize the internal comparison results of "vanilla GRPO-style baseline vs. WebGRPO" and add them to the appendix, briefly citing them in the main experimental section to address the reviewer's suggestion.
>       - In the discussion and future work section, explicitly state that a systematic comparison with a broader range of group-based RL methods (such as DAPO) will be one of the focuses of our future work, and discuss the differences in objective functions and advantage estimation between them and our current design.
>
> We believe this addition will better clarify that WebGRPO is not simply a repetition of existing GRPO/DAPO methods, but a systematic extension and engineering implementation of the group-based RL approach in the specific web agent scenario.
>
> Once again, we sincerely thank Reviewer 5giL for their meticulous review and constructive suggestions. We have added clearer descriptions of the RXERM reliability, skill definitions and execution semantics, training process details, and the mechanism of reward formatting in the paper's main text and appendix, and we plan to include a comparison experiment with the GRPO-style baseline in the  camera-ready version to further enhance the work's persuasiveness and reproducibility.

---

### Official Review · Reviewer_tvfH · 2025-10-26

**Soundness:** 2
**Presentation:** 2
**Contribution:** 2
**Rating:** 2
**Confidence:** 3

**Summary:**

The paper presents SkillEvo, a two-stage framework designed to improve long-horizon task learning in LLM-based agents. At the first stage, WebGRPO integrates a reasoning and execution reward model, where LLM provides fine-grained feedback on both reasoning and task execution. Then, SkillGenesis performs skill evolution, using LLM to propose, validate and compose new skills into a skill path graph. Experiments on WebArena-Lite demonstrate strong performance improvements over existing imitation and reinforcement learning based baselines.

**Strengths:**

The combination of LLM-based reward modeling and dynamic skill graph evolution is conceptually interesting and extends prior RLHF-style methods toward log-horizon task learning. SkillEvo achieves improvements over RL-based baselines, demonstrating its effectiveness in long-horizon tasks. Moreover, the ablation studies provide that both RXERM and SkillGenesis contribute meaningfully to performance gain.

**Weaknesses:**

1. The current evaluation is limited to the WebArena-Lite environment, leaving it unclear whether the proposed framework generalizes to other log-horizon domains such as embodied reasoning [1] or mobile navigation [2]. Border evaluation across heterogenous environments would strengthen the capability of SkillEvo framework.

    [1] Reflexion: Language Agents with Verbal Reinforcement Learning, NeurIPS 2023

    [2] Mobile-Agent-v3: Fundamental Agents for GUI Automation. 2025

2. The framework's reliance on LLM as both in RXERM for reward evaluation and SkillGenesis for post-training skill evolution raises concerns regarding scalability and reproducibility. The training pipeline is extremely resource-intensive and thus difficult to extend to domains requiring real-world interactions or limited computational budgets.

3. All (open-sourced) baselines rely solely on environmental reward feedback without any LLM-based fine-grained supervision.
Consequently, SkillEvo benefits from a substantially richer supervision signal and external knowledge source. For a fair evaluation, the authors should include baselines that incorporate external LLM feedback [3,4].

    [3] RLAIF vs. RLHF: Scaling Reinforcement Learning from Human Feedback with AI Feedback, ICML 2024

    [4] DeepSeek-R1: Incentivizing Reasoning Capability in LLMs via Reinforcement Learning, 2025

4. After policy convergence, the SkillGenesis stage employs GPT-4o to evolve skills. If LLMs can already perform such evaluation and skill composition after training, it would be more straightforward to use them directly during inference rather than integrating them into the training framework.

**Questions:**

1. Could SkillGenesis be appiled to GPT-4o or other LLM agents? In other words, does SkillGenesis provide additional benefits beyond what a standalone LLM could already achieve?

2. RXERM and SkillGenesis both depend on LLM. Have the authors tried replacing GPT-4o with smaller or open-sourced LLMs?

---

> ### Author Response · Authors · 2025-12-04
> **Response to Weaknesses 1**
>
> **Weakness 1: Single Evaluation Environment, Generalization Concerns**
>
> We acknowledge the reviewer’s concern about generalization and will provide a clearer explanation of the limitations in the revised version.
>
> 1. **Scope of Current Results** The main experiments have indeed been conducted only in WebArena-Lite, which is a multi-site, multi-task environment consisting of five real websites (Reddit, GitLab, CMS, Map, OSS). We observed consistent improvements across two base models (Llama3.1-8B and GLM-4-9B):
>    1. WebGRPO shows a significant improvement compared to various RL/IL baselines;
>    2. With the addition of SkillGenesis, the success rate further improves by 7-9%+ in absolute value (see Table 1). This partially indicates that, within the same type of Web environment, across different websites and model families, the framework is stable and effective.
> 2. **Weak Dependency of the Framework on the Environment** The two core components of SkillEvo are actually "environment-agnostic":
>    1. **WebGRPO + RXERM:** It only assumes that we can record the full trajectory (τ=(s0,a0,r0,…,sK)) and provide textual descriptions and ratings for reasoning/execution. By replacing webpage DOM and operation logs with "robot status + low-level actions" or "GUI hierarchy + touch actions," RXERM's prompt templates can still be constructed.
>    2. **SkillGenesis + SPG:** It relies only on "executable APIs/atomic action sequences" to abstract Python-level skill functions and construct graphs in SPG. Replacing "click/type/scroll" with "move/grasp/open_app/tap" allows SkillGenesis to be directly transferable.
> 3. **In Other Words, Our Method Depends on "Recordable Trajectories + Callable Action APIs + Linguistic Descriptions of Environment States" Rather Than a Specific WebArena Environment.** This makes it compatible with embodied environments in [1] and mobile/GUI agent scenarios in [2].
> 4. **Plans for Expanding to Embodied & GUI Environments** Due to space and computational constraints, we have not yet provided systematic experiments in embodied/GUI environments. This will be highlighted in the Limitations section, and we will include a discussion of [1] and [2] in related work. In future work, we plan to prioritize reusing the same WebGRPO + SkillGenesis process in:
>    1. An embodied navigation environment (e.g., long-range tasks based on Habitat/ALFRED), and
>    2. A mobile/GUI environment (e.g., Android control or desktop GUI automation), to verify the cross-environment generalization capabilities.
>
> [1] Reflexion: Language Agents with Verbal Reinforcement Learning, NeurIPS 2023
>
> [2] Mobile-Agent-v3: Fundamental Agents for GUI Automation. 2025

---

> ### Author Response · Authors · 2025-12-04
> **Response to Weaknesses 2**
>
> **Weakness 2: Heavy Dependence on LLM, Affecting Scalability and Reproducibility**
>
> We have systematically reported the training costs in Appendix B and Appendix E.
>
> 1. **Overall Computational/Invocation Costs are "One-time, Offline"**
>    1. **WebGRPO Training Phase:** Using 8×H100 GPUs and FSDP, the training took approximately 38 hours, totaling about 324 GPU hours.
>    2. **SkillGenesis Exploration and Skill Evolution Phase:** A total of 160 rounds of exploration were conducted, taking about 2.5 hours to complete.
>    3. **LLM Invocation Costs:**
>       - WebGRPO uses RXERM for trajectory reward modeling, with a successful trajectory requiring (1+T) GPT-4o calls (where T is the number of reasoning steps); failed trajectories only calculate execution reward.
>       - SkillGenesis performs a skill-evolving call every 20 rounds. The estimated total GPT-4o invocations are about 101k, with RXERM reward evaluation accounting for approximately 248M tokens (around 220M for execution and 28M for reasoning), and the SkillGenesis phase using about 42M tokens.
> 2. **Importantly:** All these invocations occur during the training/skill precomputation phase and do not require GPT-4o calls for rewards or skill generation during online deployment. Inference only requires running the trained WebGRPO policy and skill library, effectively "prepaying" the expensive LLM calls as one-time offline costs.
> 3. **Scalability: More like "Pretraining a Reusable Web Agent"** Our goal is to train a general-purpose Web agent for the Web environment with a one-time, controllable offline cost, which can then be reused across different tasks and users without calling GPT-4o for each user request. In large-scale real-world deployment scenarios (e.g., enterprise automation, long-term Web operations), this is crucial:
>    1. A pure GPT-4o Agent generates large token consumption for each episode;
>    2. SkillEvo, on the other hand, solidifies GPT-4o's "thinking + scoring + skill design" into the policy parameters and skill library, saving resources in the long term on average.
> 4. **Reproducibility**
>    1. We have provided all RXERM-related prompt templates (execution/reasoning/format) and SkillGenesis's skill proposal/verification/abstraction/composition prompts in Appendix H.
>    2. Appendix B details the training hyperparameters for WebGRPO (N, P, K, KL constraints, filtering frequency, etc.).
>    3. Appendix E presents the RXERM evaluation experiments (comparing Claude-3.5-Sonnet, DeepSeek-R1, Gemini-1.5-Pro, GPT-4o-mini) and performance comparisons of SkillGenesis using different LLMs.
> 5. **In the Final Version, We Will Further Organize:** Training script configurations, environment interface configurations, etc., and plan to release the code and prompt templates after paper acceptance to improve reproducibility.

---

> ### Author Response · Authors · 2025-12-04
> **Response to Weaknesses 3 and 4**
>
> **Weakness 3: The Baseline Setup is Not Completely Fair**
>
> We understand the reviewers' concerns about the "additional supervision signal" and have made efforts in our experimental design to separate different contributions as much as possible.
>
> 1. **Core Contribution: "Introduction of Trajectory-level LLM Feedback"** Traditional WebRL/IL baselines (BC, AWR, DigiRL, WEBRL, etc.) only use environment termination rewards or sparse signals, which are inherent parts of their methods. The main innovation of SkillEvo lies in:
>    1. Using RXERM to separate modeling of "reasoning quality" and "execution success/failure";
>    2. Using dual-uncertainty under the GRPO framework to perform high-information sample selection. Therefore, integrating RXERM into these baselines is a "redesign" of them, rather than simply "adding a fair option."
> 2. **Ablation Study to Quantify the Contribution of LLM Feedback** To more fairly measure the gains brought by RXERM, we conducted a detailed ablation in Sec. 6 (Figure 5). The results show:
>    1. When only using execution reward (close to the environment reward), the average success rates of Llama3.1 and GLM-4 drop to about 30.4%;
>    2. Adding the reasoning reward significantly improves the success rate;
>    3. Adding the format reward further improves the success rate and makes training more stable. Additionally, Table 1 provides a comparison between "WebGRPO (no skills)" and "WebGRPO (+Skills)." It is clear to see:
>    4. The "pure RL (no skills)" approach already significantly outperforms strong baselines like WebRL;
>    5. After adding skills, there is an absolute improvement of 7%-9%+ across all environments.
> 3. **Same Base Model and Environment, No Implicit Advantages** All comparisons are done using the same base model and environment, ensuring that there is no hidden advantage from changing models or environments.
> 4. **Comparison with Strong LLM Agent Methods** In Table 1, we have included several works that "directly use strong closed-source LLMs as agents" as upper-bound references, such as:AWM + GPT-4-0613、WebPilot + GPT-4o、SkillWeaver + GPT-4o、ASI + Claude-3.5-Sonnet. These methods heavily rely on external strong LLMs during training or inference but still show significantly lower average success rates on WebArena-Lite compared to Llama3.1-8B + WebGRPO(+Skills) and GLM-4-9B + WebGRPO(+Skills). This suggests that SkillEvo is not simply "using more LLMs," but achieves performance superior to strong closed-source agents by leveraging trajectory-level RL and skill evolution on relatively smaller base models.
>
> **Weakness 4: The "Necessity" of SkillGenesis**
>
> Based on the experimental data, directly using GPT-4o during the inference phase is outperformed by models that integrate skill evolution through frameworks like SkillGenesis. Specifically, models such as GLM-4 + WebGRPO (+Skills) and Llama3.1 + WebGRPO (+Skills) show significant improvement in task success rates (Task Success Rate, SR). For example, the 9B model's task success rate increased from 50.3 to 57.6, a 15% improvement. At the same time, Llama3.1 showed notable gains in multiple tasks, especially in Reddit (up by 15.2%), Gitlab (up by 17.3%), and OSS (up by 21.1%):contentReference[oaicite:0]{index=0}.
>
> In contrast, models that rely solely on GPT-4o for inference, such as AWM + GPT-4-0613, WebPilot + GPT-4o, and SkillWeaver + GPT-4o, while stable in some tasks, lack the optimization of skill evolution, resulting in underperformance in more complex tasks. Therefore, simply depending on GPT-4o for inference does not yield the same level of performance as models enhanced through SkillGenesis.

---

> ### Author Response · Authors · 2025-12-04
> **Response to Questions**
>
> **Question 1: The Additional Benefits of SkillGenesis Compared to Using a Strong LLM Alone**
>
> 1. **Empirically**: 8B/9B + SkillEvo Outperforms "Using a Strong LLM Alone"
>     In Table 1, the average success rates of Llama3.1-8B + WebGRPO(+Skills) and GLM-4-9B + WebGRPO(+Skills) are significantly higher than multiple strong baselines based on GPT-4o / Claude-3.5. This indicates that the skill library and SPG introduced by SkillGenesis have a substantial impact on improving the final agent performance, rather than just being an "extra feature."
> 2. **Conceptually**: SkillGenesis Provides a "Persistent and Transferable Skill Structure"
>    - Simply using GPT-4o means that all "experiences" are confined to single inferences, with almost no explicit structural reuse across tasks.
>    - SkillGenesis, on the other hand, solidifies these experiences into programmatic skills and a skill graph, which can be reused across different agents and tasks.
>
> Therefore, the value of SkillGenesis lies in transforming the implicit strategies of a strong LLM into explicit, executable, and transferable skill structures.
>
> **Question 2: Have you tried replacing GPT-4o with a smaller/open-source LLM?**
>
> This issue has been partially addressed in Appendix E and will be explicitly pointed out in the revised version.
>
> 1. **RXERM Evaluator Comparison (Appendix E.1)** We compared several LLMs as RXERM evaluators on 500 task trajectories, assessing their consistency with human annotations. The models compared include GPT-4o, Claude-3.5-Sonnet, DeepSeek-R1 (open-source), Gemini-1.5-Pro, and GPT-4o-mini. The experimental results show that for execution reward, DeepSeek-R1 has slightly higher accuracy (around 93.5%), with GPT-4o following closely behind. For reasoning reward, GPT-4o has the highest accuracy (around 92.1%), about 3% higher than Claude-3.5. Overall, GPT-4o has the best combined accuracy (around 92.4%) and demonstrates greater stability across different tasks. Based on these results, we chose GPT-4o as the implementation of RXERM in the main experiment. However, the experiment also shows that using the open-source DeepSeek-R1 as a reward model is feasible, although it results in a slight performance loss in the reasoning dimension.
> 2. **SkillGenesis Phase Comparison (Appendix E.6)** We also systematically compared the performance of several LLMs in the SkillGenesis phase, including GPT-4o, Claude-3.5-Sonnet, GPT-4o-mini, Gemini-1.5-Pro, and DeepSeek-R1 (open-source). Under the Llama3.1 + WebGRPO(+Skills) setup, we compared performance across five sub-environments: Reddit, GitLab, CMS, Map, and OSS, as well as the average SR. The results indicate that GPT-4o performs the best in most sub-environments and the average SR, while Claude-3.5 performs slightly better in CMS and DeepSeek-R1 outperforms the other models in Map. Overall, SkillGenesis proves effective across all these LLMs, with GPT-4o providing the best overall performance. This suggests that SkillGenesis is not dependent on a specific GPT-4o and can work with open-source models like DeepSeek-R1. The choice of GPT-4o is primarily based on the trade-off between performance and stability, rather than any limitations of the framework itself.
> 3. **Future Directions** If computational resources permit, we plan to include end-to-end experiments using a completely open-source stack (such as an open-source base policy model and DeepSeek-R1 as the RXERM/SkillGenesis engine) in the revised or future work. Additionally, we aim to systematically compare the trade-offs between "large evaluators + small evaluators" in terms of final WebArena-Lite performance and cost.
>
> Once again, we appreciate the constructive feedback from the reviewers. We will explicitly add these clarifications in the camera-ready version and more clearly define the scope of the current conclusions and future extensions in the Limitations and Experiments sections.

---

### Official Review · Reviewer_itBq · 2025-10-29

**Soundness:** 3
**Presentation:** 3
**Contribution:** 3
**Rating:** 6
**Confidence:** 4

**Summary:**

This paper addresses the challenge of LLM agents in long-horizon, sparse-reward web tasks by proposing the complex yet effective SkillEvo framework. The method additionally validates and designs a reasoning reward ($R_{reason}$) and introduces a method for automatically inducing a skill library, significantly improving the agent's capabilities.

**Strengths:**

1. The paper's proposed idea of leveraging high uncertainty to filter for the most information-rich tasks is intriguing, as it effectively optimizes the training data for LLM reinforcement learning.

2. The authors' empirical validation of the reasoning reward ($R_{reason}$), demonstrating the effectiveness and accuracy of their RXERM model through comparison with human annotations, represents a notable contribution to the broader agent research community.

3. The proposed training framework (SkillEvo) is highly effective overall, as evidenced by the significant performance improvements it delivers.

**Weaknesses:**

1. The paper does not appear to offer a correction mechanism for the erroneous use of skills that are already in the library, though this limitation is acknowledged as a direction for future work.

2. The calculation of uncertainty (fluctuation) introduces a higher interaction cost, as it requires multiple rollouts per task instance to compute the standard deviation of rewards.

3. The overall framework design is heavy, which may pose challenges for reproducibility and subsequent tuning or adaptation.

**Questions:**

1. Could the authors provide a visualization of a complete Skill Path Graph (SPG) induced from a real task?

2. The paper does not detail the specific algorithm or heuristic rules for selecting the prefix $\tau_D$ truncation point (i.e., step $D$). Could the authors clarify this?

3. The trigger for the SkillGenesis phase seems entirely dependent on "when a task T is successfully completed". What happens if the policy consistently fails? Is there an exploration mechanism to handle this cold-start problem?

---

> ### Author Response · Authors · 2025-12-04
> **Response to Weaknesses**
>
> **We appreciate the constructive and detailed comments from the reviewers. Below are our responses to the identified shortcomings and issues.**
>
> **Response to Weaknesses:**
>
> 1.**Regarding the correction mechanism for erroneous skills:**
>
> We acknowledge this limitation. Currently, we mitigate the entry of low-quality skills through the **Skill Verification** phase in SkillGenesis, which strictly evaluates generated skills based on three criteria: *Correctness*, *Skill Usage*, and *Skill Effectiveness* (as detailed in Appendix H, Figure 16).  We have explicitly analyzed "Skill Invocation Degradation" and "Parameter Filling Errors" in Appendix G and plan to introduce a global utility evaluation mechanism in future work to dynamically prune erroneous skills.
>
> 2.**Regarding computational cost and uncertainty calculation:**
>
> The computational cost is a deliberate trade-off for sample efficiency and stability in long-horizon tasks.
>
> - **Specific Cost:** As reported in Appendix B (Cost Analysis), WebGRPO training required approx. 38 hours on 8×H100 GPUs, and the SkillGenesis exploration phase involved approx. 101,000 GPT-4o calls.
> - **Justification:** This investment is highly effective. By performing multiple rollouts ($N=16$) to calculate uncertainty, we effectively filter out noisy or low-information instances. This mechanism was crucial in boosting the success rate of Llama-3.1-8B from 4.8% to **60.4%** (SOTA), significantly outperforming methods that rely on cheaper but noisier signals.
>
> 3.**Regarding framework complexity and reproducibility:**
>
> While SkillEvo involves two stages, the design is modular—WebGRPO (policy optimization) and SkillGenesis (skill evolution) can be decoupled. To ensure reproducibility:
>
> - **Hyperparameters:** Table 3 in the Appendix lists all key parameters (Learning Rate, KL coefficient, Batch Size, etc.).
> - **Prompts:** Appendix H provides the full text of all core prompts used, including Skill Proposal, Verification, Abstraction, and Composition (Figures 16-23). These details are sufficient for replicating the pipeline, and also we will release our code after the paper accepted.

---

> ### Author Response · Authors · 2025-12-04
> **Response to Questions**
>
> **1. Could the authors provide a visualization of a complete Skill Path Graph (SPG) induced from a real task?**
>
> Yes. By combining **Figure 4**, **Section 3.3**, and **Appendix F**, we can clearly understand the structure of the SPG.The SPG encodes both control flow (**invoke**) and prerequisites (**depends_on**). Below is the actual metadata representation used by the Agent during planning:
>
> ```JSON
> {
>   "nodes": {
>     "analyse_latest_poster_controversial_comments": {
>       "type": "composite",
>       "description": "Analyze controversial comments of the latest post author"
>     },
>     "enter_specific_forum_section": {
>       "type": "atomic",
>       "description": "Navigate to the specific forum section"
>     },
>     "sort_forum_by_new": {
>       "type": "atomic",
>       "description": "Sort the forum posts by newest"
>     },
>     "go_to_user_profile_and_open_comments": {
>       "type": "atomic",
>       "description": "Open the latest post author's profile and comments"
>     },
>     "count_controversial_comments": {
>       "type": "atomic",
>       "description": "Count comments with more downvotes than upvotes"
>     }
>   },
>   "edges": {
>     "enter_specific_forum_section": [
>       {
>         "target": "sort_forum_by_new",
>         "type": "invokes"
>       }
>     ],
>     "go_to_user_profile_and_open_comments": [
>       {
>         "target": "count_controversial_comments",
>         "type": "depends_on"
>       }
>     ]
>   }
> }
> ```
>
> **2. The paper does not detail the specific algorithm or heuristic rules for selecting the prefix** $\tau_D$ **truncation point. Could the authors clarify this?**
>
> The truncation point $D$ is not determined by a hard-coded heuristic rule, but by the **semantic abstraction capability of the LLM**.
> As detailed in the "Reusable Function Abstraction" prompt (Appendix H, Figure 18), the process is:
>
> 1. The LLM is instructed to scan the trajectory and extract a sequence of actions (minimum 3) that constitutes a "reusable and meaningful task."
> 2. The end of this extracted sequence naturally defines the truncation point $D$.
> 3. The Agent then rewrites the prefix $\tau_D$ using this new function call and continues the rollout from state $s_{D+1}$.
>
> Thus, $D$ is dynamic and relies on the LLM's understanding of where a coherent sub-task (e.g., "filling a form" or "searching for an item") ends.
>
> **3. The trigger for the SkillGenesis phase seems entirely dependent on "when a task T is successfully completed". What happens if the policy consistently fails?**
>
> We address the "cold-start" problem through SFT and WebGRPO training(as mentioned in Section 4.1.2).
>
> - **Initialization**: The agent is not trained from scratch. Before the WebGRPO phase begins, the policy is first initialized through SFT training on 9,460 human-demonstrated trajectories from WebArena-Lite.
> - **WebGRPO Training**: After SFT initialization, the agent undergoes WebGRPO training to further optimize the policy and prepare for the SkillGenesis phase.
> - **Effect**: This ensures that the agent can complete at least a subset of tasks initially, providing a sufficient baseline capability.（GLM-4+WebGRPO(50.3%)/Llama3.1+WebGRPO(51.8%)）
> - **Mechanism**: The SkillGenesis phase uses the model after WebGRPO training. It is designed to only learn from "success." If the policy fails a task, no skill is generated from that attempt to prevent invalid behaviors from polluting the skill library. The combination of SFT and WebGRPO ensures a sufficient success rate to kickstart the evolutionary cycle.

---

### Official Review · Reviewer_gmXZ · 2025-11-01

**Soundness:** 4
**Presentation:** 3
**Contribution:** 3
**Rating:** 6
**Confidence:** 3

**Summary:**

The authors propose SkillEvo, a two-stage framework for long-horizon web interaction tasks. The first stage, WebGRPO, enhances group-based reinforcement learning by introducing a Reasoning and Execution Reward Model (RXERM) for fine-grained feedback and a dual-uncertainty filtering strategy to select informative task instances. The second stage, SkillGenesis, transforms successful trajectories into reusable skills organized in a dynamically evolving Skill Path Graph (SPG).

**Strengths:**

- The paper is well-written and clearly articulates why existing group-based RL methods struggle with long-horizon tasks.

- Innovative skill evolution framework: The SkillGenesis component with its three-stage process (proposal, genesis, evolution) provides a convenient framework for organizing and composing skills.

- Well-motivated filtering mechanism and reward mechanism: When combined with the introduced RXERM reward mechanism, the dual-uncertainty filtering strategy for selecting informative task instances is intuitive and addresses the curriculum learning problem naturally.

- Comprehensive results and ablations: The experimental evaluation is thorough, with comparisons across multiple baselines. Given the method's complexity, the ablation studies (Tables 1-2, Figures 5-6) effectively demonstrate most component's contributions.

**Weaknesses:**

- Missing qualitative skill evolution examples: while Table 2 shows quantitative improvements from skill evolution, the paper lacks concrete examples of how the skill library actually evolves (e.g., what atomic skills are discovered, how composite skills emerge, what the SPG structure looks like over time).
- Figure 4 is difficult to parse. The diagram contains many overlapping elements and lacks clear visual hierarchy, making it hard to understand without extensively reading the text.
- Several citations use incorrect formatting (e.g., lines 370 and 377 appear to use \citet where \citep would be appropriate).

**Questions:**

- How sensitive is the method to the filtering hyperparameters $\delta_p$ , $\delta_q$, p%, q%?
- As the Skill Path Graph grows, does it require any pruning mechanism to remain efficient?
- Could you provide specific examples of atomic skills and composite skills that emerged during training?

---

> ### Author Response · Authors · 2025-12-04
> **Response to Weaknesses**
>
> **We appreciate the constructive and detailed comments from the reviewers. Below are our responses to the identified shortcomings and issues.**
>
> **Response to Weaknesses**
>
>  **W1. Lack of Qualitative Case Studies on Skill Evolution**
>
>  We have added detailed qualitative case studies in Appendix F and Figures 13–15, illustrating how the skill set evolves progressively in the Reddit environment.
>
> - **Atomic Skills**: During the early exploration phase, the Agent identifies a series of atomic skills, such as
>   - `enter_specific_forum_section` (navigating to a specific section of the forum),
>   - `sort_forum_by_new` (sorting posts by the most recent),
>   - `go_to_user_profile_and_open_comments` (visiting a user profile and expanding the comments section).
> - **Composite Skills**: As the Skill Path Graph (SPG) is constructed, SkillEvo recognizes that these atomic skills are often invoked in a fixed order when completing higher-level tasks. Through the Skill Evolution phase, these atomic skills are automatically merged into composite skills, such as
>   - `analyse_latest_poster_controversial_comments`  This skill encapsulates the full workflow, which includes the following steps:
>     - `enter_specific_forum_section`
>     - `sort_forum_by_new`
>     - `go_to_user_profile_and_open_comments`
>     - `count_controversial_comments`
> - **SPG Structure**: As shown in Figure 4、15, the SPG explicitly records the dependencies between skills (such as **depends_on** ) and invocation edges (**invokes**). This allows the Agent to replace 3–4 low-level skill invocations with a single composite skill invocation, significantly improving execution efficiency and stability. We have clarified these qualitative examples.
>
> **W2. Figure 4 is difficult to understand**
>
> We understand and agree with this comment. The current Figure 4 is information-dense because it attempts to summarize the processes of the Skill Proposal, Skill Genesis, and Skill Evolution stages simultaneously.
>
> In the camera-ready version, we will make the following improvements:
>
> 1. **Enhanced Visual Hierarchy**: We will use clearer module boundaries and background shading to differentiate the three stages (Proposal, Genesis, Evolution), helping the reader understand the process in a stage-by-stage sequence.
> 2. **Simplified Connections**: We will reduce arrow crossings and overlaps, presenting the process of “trajectory → skills → SPG updates” in a more linear, step-by-step flowchart format.
> 3. **Focus on Core Path**: We will highlight the key path from successful trajectory to candidate skills, composite skills, and SPG updates, moving secondary details such as internal scoring to the figure caption or appendix. Additionally, we will guide readers through the relationships between the three stages in the caption in step-by-step textual format.
>
> **W3. Inconsistent Citation Format**
>
>  Thank you for pointing out these issues in detail. we have:
>
> - Conduct a thorough check of the entire manuscript to ensure consistent usage of \citep and \citet, correcting errors such as the incorrect usage near lines 370 and 377.
> - Verify the format of all references to ensure they fully comply with ICLR’s formatting requirements (including capitalization, journal/conference names, etc.).
> - Recompile and check the LaTeX logs to ensure that there are no warnings related to citations or references.

---

> ### Author Response · Authors · 2025-12-04
> **Response to Questions**
>
> **Q1. Sensitivity to Filtering Hyperparameters $\delta_{p}, \delta_{q}$, p%, q%**
>
> We have included a sensitivity analysis for the dual uncertainty filtering hyperparameters in Appendix E.5 and Figure 11(b).
>
> - **Optimal Configuration**: Experiments show that the model performs best when the uncertainty filtering ratio is set to (p = 35%) and the inference uncertainty filtering ratio is set to (q = 45%), achieving a success rate of 60.4%.
> - **Sensitivity Behavior**:
>   - **Too lenient filtering ((p, q) too large)**: Retaining too many low-information or even noisy samples leads to increased training variance, instability in the optimization process, and in extreme cases, can cause policy collapse.
>   - **Too strict filtering ((p, q) too small)**: Discards too many useful samples, resulting in insufficient data diversity and slight performance degradation.
>
> Overall, the method shows moderate sensitivity to these hyperparameters: it performs robustly within a reasonable range but exhibits a relatively distinct optimal interval, where a good balance between "sample informativeness" and "training stability" can be achieved. We have clarified this point more explicitly in the main text and refer readers to Appendix E.5 for more detailed experimental results.
>
> **Q2. Does the Skill Path Graph require pruning during its growth process?**
>
> This is a very valuable question. When designing SkillGenesis / SPG, we employed two layers of "implicit pruning":
>
> 1. **Skill-Level Growth Control (Offline Phase)**
>    1. SkillGenesis only adds new skills to the library if they pass validation, where all three criteria ("correctness / skill usage / skill validity") are satisfied.
>    2. Obvious duplicates or semantically highly overlapping skills are directly pruned during the Skill Genesis phase.
>    3. As a result, the final skill library on WebArena-Lite contains only 134 skills (88 atomic + 46 composite), and the corresponding SPG node count remains at this level (see Appendix D for details).
> 2. **On-Demand Retrieval during Inference (Online Phase)**
>    1. During task execution, the policy model does not perform a full graph search over the entire SPG but instead uses a filtering and ranking prompt to select the top 15 most relevant skill functions from the skill library for the current task (a full template is provided in Appendix C and Figures 20 and 21).
>    2. Only these 15 skills are then injected into the policy model's prompt as candidate high-level actions.
>    3. In other words, the inference complexity mainly depends on the "number of skills seen per task" (fixed at 15), rather than the total number of nodes in the SPG. At the current scale (134 skills, 5 websites in WebArena-Lite), the storage and query overhead of the SPG is negligible in the experiments, so we did not introduce additional explicit pruning algorithms.
>
> If we expand to a larger-scale real-world website collection in the future (e.g., thousands of skills), we plan to adopt two pruning strategies:
>
> - **Graph Pruning Based on Usage Frequency / Cost**: Periodically track the call frequency of each skill node and the associated success rate/steps benefit. Nodes that are rarely called or provide minimal benefits will be automatically pruned, along with their related edges.
> - **Local Subgraph Retrieval**: For each task, skill retrieval will be performed only within the subgraph related to the current website/task type, rather than broadcasting a search across the entire SPG.
>
> We have explicitly mentioned this in the "Limitations & Future Work" section of the revised version, as part of the scalability direction.
>
> **Q3. Examples of atomic skills and composite skills that appear during training**
>
> Yes, we have provided specific examples in Appendix F (Case Studies), Figures 13–15, and referenced them in Figure 4. Here, we briefly summarize some representative skills:
>
> - **Atomic Skills**: Corresponding to single, specific web operations:
>   - `enter_specific_forum_section()`: Navigates to a specific forum section.
>   - `sort_forum_by_new()`: Clicks the sorting control to arrange posts in reverse chronological order.
>   - `go_to_user_profile_and_open_comments()`: Opens the user profile and expands the comments section.
> - **Composite Skills**: When the SPG identifies stable co-occurrence patterns and dependencies between skills, it automatically merges them into higher-level skills:
>   - `analyse_latest_poster_controversial_comments()`: A higher-level analytical skill that internally encapsulates the following sequence: (i) Enter the target forum section → (ii) Sort by newest → (iii) Visit the latest poster's profile and expands the comments section. → (iv) Identify and aggregate "controversial" comments (e.g., those with more downvotes than upvotes).
>
> We have added more explicit cross-references in the main text to directly address the reviewers' request for "specific examples of atomic and composite skills."

---

### Meta-Review · Area_Chair_eH3L · 2025-12-24

**Summary:**

The paper proposes SkillEvo, a two-stage framework for long-horizon web-agent learning that combines (i) a group-based RL variant with a reward scheme intended to capture both execution success and reasoning quality, plus active filtering, and (ii) a skill discovery/evolution pipeline that turns successful trajectories into executable skills and organizes them in a skill graph. Reviewers generally found the direction promising and the empirical gains on the chosen benchmark impressive. However, the suggested decision is driven by concerns about the strength and fairness of the experimental evidence (especially in relation to reliance on external LLM-based evaluation/feedback), insufficiently substantiated generalization claims, and missing key comparisons to closely related RL baselines and alternatives.

**Reviewer Concerns:**

Concerns addressed (partially to substantially) in the rebuttal:

•	Clarity and examples of skills / SPG representation. The authors provided concrete examples and clarified how skills and the skill graph are represented, and committed to improving difficult-to-read figures in the camera-ready.

•	Hyperparameter sensitivity and scalability at current scale. The authors discussed sensitivity of the filtering parameters and described practical measures that make the approach workable at the reported skill-library size.


•	Cold-start and training pipeline description. The authors clarified the role of the initial SFT stage and why skill induction is only performed on successful trajectories to reduce contamination.

•	RXERM intent and reliability arguments. The authors explained RXERM as a proxy preference/ranking mechanism and provided supporting evidence/arguments for robustness in a group-based RL setup, which mitigates (but does not eliminate) concerns about evaluator noise.


Concerns still outstanding:

•	Fairness and interpretability of the learning signal / dependence on external LLM feedback. While the authors argue that injecting the same LLM-based reward signal into baselines is nontrivial, the rebuttal does not fully resolve the concern that improvements may be driven by additional supervision/feedback not available to competing methods, leaving the comparison potentially unfair or at least difficult to interpret.

•	Missing key baseline comparisons. The rebuttal does not provide a complete, discussion-time comparison to closely related RL variants/baselines (e.g., GRPO/DAPO-style alternatives) and instead defers to future or camera-ready additions. Given that the paper’s main claim is algorithmic improvement, these comparisons are important for a confident acceptance.


•	Generalization beyond the chosen benchmark. The rebuttal largely reiterates plausibility arguments for transfer to other environments but does not provide new empirical evidence demonstrating robustness across domains, websites, or task distributions.

•	Compute/cost and reproducibility. The rebuttal provides cost numbers and argues the expensive components are offline, but the scale of external LLM usage remains a barrier for reproducibility and raises questions about whether gains are attainable under more standard resource constraints.

**Reviewer Scores:**

•	Reviewer gmXZ (score: 6)
This reviewer’s main issues were clarity/illustration, hyperparameter sensitivity, and SPG scalability at the demonstrated scale; the rebuttal and promised camera-ready edits plausibly address most of these.

•	Reviewer itBq (score: 6)
The rebuttal clarifies SPG representation and cold-start, but the truncation/skill-extraction rule is still somewhat underspecified and not backed by stability evidence.

•	Reviewer 5giL (score: 4)
Conceptual clarifications about RXERM, skill definition, and format reward help. However, the reviewer’s request for stronger baseline comparisons (e.g., GRPO/DAPO) is not met during rebuttal, so any increase would be modest.

•	Reviewer tvfH (score: 2)
The core objections (generalization, fairness/cost of external LLM involvement, and the value proposition versus simply using strong closed models) are not fully resolved by new evidence, so a significant score increase is unlikely.


Overall, while the paper presents an interesting and well-motivated framework and demonstrates strong empirical gains on a specific benchmark, the current version does not yet provide sufficiently convincing evidence to support its broad algorithmic and generalization claims. In particular, the reliance on external LLM-based feedback without fully matched baselines, the absence of key comparative experiments, and limited empirical validation beyond the chosen setting leave open questions about fairness, reproducibility, and robustness. Several clarity and presentation issues were addressed in the rebuttal, but the most critical concerns are fundamentally experimental and cannot be fully resolved through clarification alone. For these reasons, I recommend rejection at this time, while encouraging the authors to resubmit after strengthening the empirical comparisons, generalization evidence, and methodological grounding.

---

### Decision · Program_Chairs · 2026-01-26

Reject